# How to *Correctly* Report LLM-as-a-Judge Evaluations

**Chungpa Lee**[† 1] **Thomas Zeng**[2] **Jongwon Jeong**[2] **Jy-yong Sohn**[1] **Kangwook Lee**[2 3 4]

## Abstract

Large language models (LLMs) are widely used as scalable evaluators of model responses in lieu of human annotators. However, imperfect sensitivity and specificity of the LLM judges induce bias in naive evaluation scores. We propose a simple plug-in framework that corrects this bias and enables statistically principled uncertainty quantification. Our framework constructs confidence intervals that account for uncertainty from both the test dataset and a human-labeled calibration dataset. Additionally, it uses an adaptive strategy to allocate calibration samples for tighter intervals. Importantly, we characterize parameter regimes defined by the true evaluation score and the LLM judge's sensitivity and specificity in which our LLM-based evaluation yields more reliable estimates than human-only evaluation. Moreover, we show that our framework remains unbiased under distribution shift between the test and calibration datasets, in contrast to existing approaches.

## 1. Introduction

The use of large language models (LLMs) as judges has become a low-cost and scalable alternative to human evaluation across various tasks (Zheng et al., 2023; Liu et al., 2023; Wang et al., 2023; Li et al., 2025; Gu et al., 2024). A common practice is to summarize such evaluations by the fraction of responses that the LLM judges as *'correct'*, denoted by $\hat{p}$. However, this seemingly simple reporting practice is statistically problematic (Bross, 1954; Schwartz, 1985; Forman, 2005; Angelopoulos et al., 2023a; Boyeau et al., 2025; Fraser, 2024; Albinet, 2025). Because LLM judgments are noisy, the raw score $\hat{p}$ generally differs from the true accuracy and can lead to unreliable conclusions (Wang et al., 2024; Koo et al., 2024; Huang et al., 2025).

[†]Work done while visiting the University of Wisconsin–Madison. [1]Yonsei University, Korea [2]University of Wisconsin–Madison, USA [3]KRAFTON, Korea [4]Ludo Robotics, Korea. Correspondence to: Jy-yong Sohn <jysohn1108@yonsei.ac.kr>, Kangwook Lee <kangwooklee@krafton.com>.

*Proceedings of the $43^{rd}$ International Conference on Machine Learning*, Seoul, South Korea. PMLR 306, 2026. Copyright 2026 by the author(s).

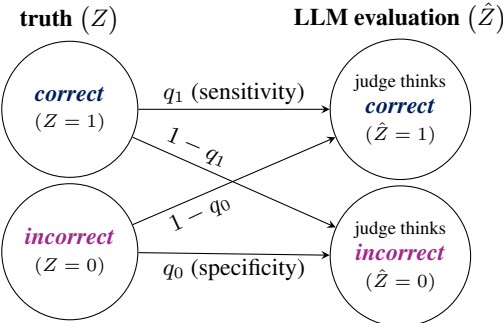

*Figure 1.* LLM judgments with error rates $1-q_1$ and $1-q_0$, where $q_1$ and $q_0$ are the LLM's sensitivity and specificity, respectively.

To understand the source of bias in the raw judgment score $\hat{p}$, we first examine how an LLM judge can make errors on individual responses. As illustrated in Figure 1, an LLM may incorrectly label an *'incorrect'* response as *'correct'*, or conversely, mislabel a *'correct'* response as *'incorrect'*. Let $q_1$ and $q_0$ denote the probabilities that the LLM correctly judges *'correct'* and *'incorrect'* responses, respectively. These quantities correspond to the sensitivity $q_1$ and specificity $q_0$ of the LLM judge.

The bias can be seen even in a simple extreme case. Suppose that $q_1 = 1$ and $q_0 = 0$. Then the LLM judges every response as *'correct'*, so the raw LLM judgment score satisfies $\hat{p} = 1$ regardless of the true accuracy $\theta$.

In general, whenever the LLM is imperfect ($q_0 + q_1 < 2$), the expected value of $\hat{p}$ deviates from the true accuracy $\theta$:

$$\mathbb{E}[\hat{p}] = \theta + (1 - q_0 + 1 - q_1)\left(\tfrac{1-q_0}{1-q_0+1-q_1} - \theta\right).$$

This expression shows that $\hat{p}$ has positive bias at low values of $\theta$ and negative bias at high values of $\theta$; see Section 5 for details. Figure 2a illustrates this behavior for an LLM judge with $1 - q_0 = 0.3$ and $1 - q_1 = 0.1$. In this case, $\mathbb{E}[\hat{p}]$ overestimates $\theta$ when $\theta < 0.75$ (blue line) and underestimates it when $\theta > 0.75$ (red line). These two directions of bias arise from the two types of judgment errors (green arrows). A high probability of incorrectly rejecting a *'correct'* response, corresponding to a large $1 - q_1$, induces negative bias at high accuracies. Conversely, a high probability of incorrectly accepting an *'incorrect'* response, corresponding to a large $1 - q_0$, induces positive bias at low accuracies.

This issue is not merely theoretical. As LLM-based evaluation becomes more widely used, reported improvements

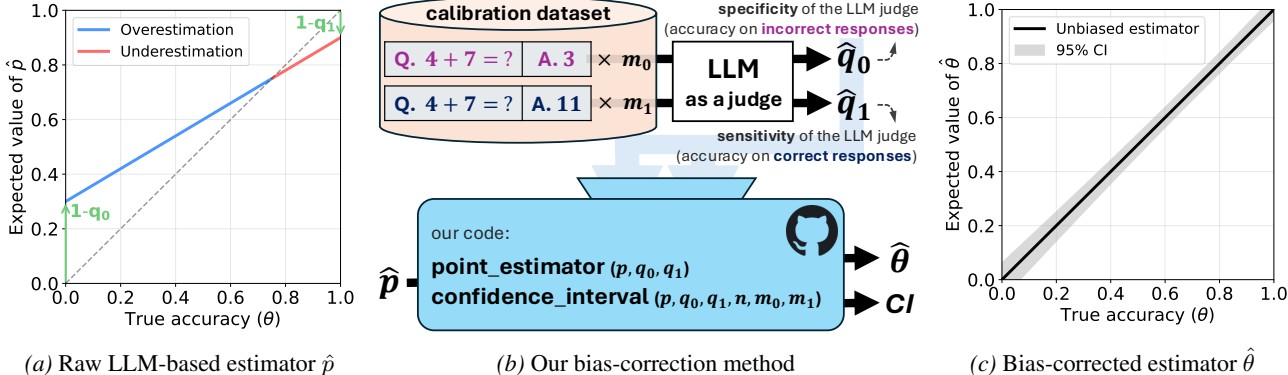

*(a)* Raw LLM-based estimator $\hat{p}$      *(b)* Our bias-correction method      *(c)* Bias-corrected estimator $\hat{\theta}$

*Figure 2.* Bias and its adjustment in LLM-based judgment under imperfect LLM evaluators ($1 - q_0 = 0.3$ and $1 - q_1 = 0.1$). *(a)* When the true accuracy $\theta$ is low ($\theta < 0.75$), the expected value of the raw LLM-based estimator $\mathbb{E}[\hat{p}]$ overestimates $\theta$, whereas when $\theta$ is high ($\theta > 0.75$), it underestimates $\theta$. *(b)* By accounting for the LLM judge's sensitivity $q_1$ and specificity $q_0$, we obtain the bias-corrected estimator $\hat{\theta}$ and its confidence interval (CI). We estimate $q_1$ and $q_0$ using a calibration dataset with true labels from human evaluators paired with LLM judgments. Here, $n$, $m_0$, and $m_1$ denote the test-set size, the number of calibration samples whose true label is incorrect (e.g., $4 + 7 = 3$), and the number of calibration samples whose true label is correct (e.g., $4 + 7 = 11$), respectively. *(c)* The resulting estimator $\hat{\theta}$ is unbiased when the true values of $q_0$ and $q_1$ are known, or when a sufficiently large calibration dataset is available. A plug-in Python implementation of this procedure is provided at `https://github.com/UW-Madison-Lee-Lab/LLM-judge-reporting`.

may reflect bias from judgment errors rather than genuine model gains. Since different evaluation procedures can induce biases of different magnitudes and directions, some apparent advances may arise from evaluation artifacts. This motivates careful comparison of LLM-based evaluation results and highlights the need for bias adjustment.

Fortunately, this bias can be corrected. When $q_1$ and $q_0$ are known, a classical result (Rogan & Gladen, 1978) provides an exact adjustment. When they are unknown, they can be estimated from a calibration dataset with human-evaluated labels, and the resulting estimates $\hat{q}_1$ and $\hat{q}_0$ can be substituted into the correction formula. This leads to the bias-corrected estimator $\hat{\theta}$ illustrated in Figure 2c.

More broadly, several bias-correction methods have been studied for imperfect evaluators. In particular, prediction-powered inference (Angelopoulos et al., 2023a;b) provides a general framework that extends beyond categorical outcomes and can yield estimators with lower variance; see also Kloos et al. (2021); Chen et al. (2026). However, we show that existing methods can become biased under distribution shift between the test and calibration datasets, a setting that can arise in LLM-as-a-judge evaluation (Jung et al., 2024). In contrast, the estimator $\hat{\theta}$ based on Rogan & Gladen (1978) remains unbiased under such shifts. We therefore adopt this estimator and focus our analysis on $\hat{\theta}$. Since $\hat{\theta}$ can have higher variance than some existing alternatives, we further propose an adaptive strategy for allocating calibration samples, which reduces its variance and shortens the corresponding confidence interval.

In this paper, we provide a statistical framework for correctly reporting LLM-as-a-judge evaluations, as outlined in Figure 2b. We summarize our key contributions below:

- In Section 4, we introduce the bias-corrected estimator for the true accuracy $\theta$, derive the confidence interval, and further propose an adaptive strategy for allocating calibration samples that reduces its length. In Section 5, we establish theoretical guarantees for our method.

- In Section 6, we characterize parameter regimes defined by $q_1$, $q_0$, and $\theta$ in which our estimator has smaller variance than that obtained from human-only evaluation. We also examine where contemporary LLM-as-a-judge approaches fall within these regimes.

- In Section 7, we validate our method through extensive Monte Carlo simulations and real-world LLM-as-a-judge evaluations on the Chatbot Arena benchmark.

- In Section 8, we show that estimators from existing bias-correction methods can still be biased under distribution shift between the test and calibration datasets, whereas our estimator remains unbiased under such shifts.

## 2. Related Work

**Statistical Reliability in LLMs' Evaluation.** Recent efforts have sought to formalize the statistical reliability of language model evaluations. While frameworks for calculating error bars and confidence intervals in benchmarks exist (Miller, 2024), they assume that LLM evaluators provide true labels. However, this assumption does not hold in the LLM-as-a-judge setting, where imperfect judges introduce bias into LLM-based scores. Recent works have examined these biases across various contexts, including natural language inference and test-time compute scaling (Godbole & Jia, 2025; Mukherjee et al., 2025; Feng et al., 2025). Aligned with this line of work, we propose a bias-corrected estimator together with statistically sound confidence intervals.

**Estimation Methods to Mitigate Bias.** To address bias arising from imperfect evaluators (e.g., diagnostic or screening tests, or LLM-based judges), Rogan & Gladen (1978) proposed an adjustment that corrects prevalence estimates when sensitivity and specificity are known. Subsequent work has developed more general frameworks, most notably Prediction-Powered Inference (Angelopoulos et al., 2023a;b; Zrnic & Candès, 2024; Broska et al., 2025; Boyeau et al., 2025), which leverages a small set of true labels to improve estimation, as well as conditional calibration estimators (Buonaccorsi, 2010; Kloos et al., 2021; Meertens et al., 2022); see Section 8 for details. While these methods offer distinct advantages, such as variance reduction (Kloos et al., 2021; Chen et al., 2026), our approach builds on Rogan & Gladen (1978); Lang & Reiczigel (2014), adopting distributional assumptions that remain valid under shifts between the calibration and test datasets and are particularly well-suited to the LLM-as-a-judge setting.

## 3. Problem Setup: LLM-as-a-Judge

We consider the problem of evaluating responses based on human judgment. For example, each instance consists of a question and a corresponding response produced by a given model.[1] We focus on binary evaluations, where each response is judged as either *'correct'* or *'incorrect'*. In Appendix B, we discuss an extension to evaluations with more than two categories, such as grading or rating-scale evaluation.

We assume that humans can assess whether a response is *'correct'* or *'incorrect'*.[2] This assessment is formalized by a ground-truth labeling function $z : \mathcal{X} \to \{0, 1\}$, where $z(x) = 1$ indicates that humans judge the response in instance $x$ to be *'correct'*, and $z(x) = 0$ otherwise. Applying the function to a random instance $X$ induces a binary random variable $Z = z(X)$.

Our goal is to estimate the true accuracy of the responses with respect to human judgment, defined as

$$\theta := \Pr(Z = 1) = \mathbb{E}[Z]. \tag{1}$$

**Test Distribution and LLM-Based Judgment.** Let $\mathbb{P}$ denote the distribution over test instances to be evaluated. In practice, instead of relying on human annotators, an LLM $f_{\mathrm{LLM}}$ is used as a judge. Let $\hat{Z} := f_{\mathrm{LLM}}(X) \in \{0, 1\}$ denote the LLM's judgment, where $\hat{Z} = 1$ indicates that the LLM marks the response as *'correct'*, and $\hat{Z} = 0$ otherwise.

---

[1]The model producing the response may be an LLM, but rule-based or statistical models are also possible.

[2]We assume that human disagreement is absent or resolved by a predefined protocol, and treat human evaluation as the ground truth. Extensions that explicitly model human disagreement are discussed as future work in Appendix A.

Let $[n] := \{1, \cdots, n\}$, where $n$ denotes the test-set size. Given a test dataset $\{x_i\}_{i \in [n]}$ sampled independently and identically from $\mathbb{P}$, the LLM produces predictions $\hat{z}_i := f_{\mathrm{LLM}}(x_i)$. The quantity reported in practice is the empirical fraction of instances judged as *'correct'* by the LLM:

$$\hat{p} := \frac{1}{n} \sum_{i \in [n]} \hat{z}_i. \tag{2}$$

This estimator targets the probability $p := \Pr(\hat{Z} = 1)$ that the LLM judges a test instance as *'correct'*.

However, the LLM's judgment $\hat{Z}$ does not necessarily coincide with the human-evaluated true label $Z$. We characterize the LLM judge using the following parameters:

$$q_1 := \Pr(\hat{Z} = 1 \,|\, Z = 1), \ q_0 := \Pr(\hat{Z} = 0 \,|\, Z = 0), \tag{3}$$

which correspond to the *sensitivity* (true positive rate) and *specificity* (true negative rate) of the LLM judge, respectively (Forman, 2008; Lang & Reiczigel, 2014). Consequently, the LLM may incorrectly reject responses that are truly *'correct'* with probability $1 - q_1$, or accept responses that are truly *'incorrect'* with probability $1 - q_0$. Unless the LLM is perfectly accurate (i.e., $q_0 = q_1 = 1$), the naive estimator $\hat{p}$ in (2) is generally a biased estimator of $\theta$.

**Calibration Distribution and Estimation of Sensitivity and Specificity.** Let $\mathbb{Q}$ denote the distribution over calibration instances. Unlike the test dataset, each calibration instance is evaluated by both humans and the LLM judge, so that both the true label $Z$ and the corresponding LLM judgment $\hat{Z}$ are observable.

Let $m$ denote the calibration-set size, and let $m_1$ and $m_0$ denote the numbers of calibration instances with $z_j = 1$ and $z_j = 0$, respectively, so that $m = m_0 + m_1$. The index $j$ distinguishes calibration instances from test instances indexed by $i$. Using the calibration dataset, we estimate the sensitivity $q_1$ and specificity $q_0$ of the LLM judge in (3) as

$$\hat{q}_1 := \frac{\sum_{j \in [m]} \mathbf{1}\{\hat{z}_j = 1, z_j = 1\}}{m_1}, \ \hat{q}_0 := \frac{\sum_{j \in [m]} \mathbf{1}\{\hat{z}_j = 0, z_j = 0\}}{m_0},$$

where $\mathbf{1}\{\cdot\}$ denotes the indicator function. In the main analysis, we assume $\mathbb{P} = \mathbb{Q}$. In Section 8, we relax this assumption and consider distribution shift, where $\mathbb{P} \neq \mathbb{Q}$.

**Problem Statement.** Because the LLM judge is imperfect, the naive estimator $\hat{p}$ is generally biased for the true accuracy $\theta$ defined in (1), i.e., $\mathbb{E}[\hat{p}] \neq \theta$. Moreover, existing LLM-as-a-judge evaluations typically report this point estimate without quantifying uncertainty via confidence intervals.

Our objective is therefore twofold: (i) to construct a bias-corrected estimator of the true accuracy $\theta$, and (ii) to provide statistically sound confidence intervals that reflect uncertainty arising from both the test and calibration datasets.

# 4. Method to Correctly Report LLM-as-a-Judge Evaluations

In this section, we present a bias-corrected estimator $\hat{\theta}$ and a confidence interval for LLM-as-a-judge evaluations. We further propose an adaptive allocation strategy for constructing the calibration dataset that reduces the confidence interval length. Derivations and theoretical guarantees for the proposed method are provided in Section 5.

## 4.1. Mitigating Bias in Point Estimator

We begin with the setting in which the sensitivity $q_1$ and specificity $q_0$ in (3) are known. In this case, an unbiased estimator of the true accuracy $\theta$ in (1) is given by

$$\hat{\theta} \mid q_0, q_1 = \frac{\hat{p} + q_0 - 1}{q_0 + q_1 - 1}. \qquad (4)$$

In realistic settings, these parameters $q_1$ and $q_0$ are unknown and must be estimated from a calibration dataset. Substituting estimates $\hat{q}_1$ and $\hat{q}_0$ into (4) gives the bias-corrected estimator (Rogan & Gladen, 1978; Lang & Reiczigel, 2014):

$$\boxed{\hat{\theta} = \frac{\hat{p} + \hat{q}_0 - 1}{\hat{q}_0 + \hat{q}_1 - 1}.} \qquad (5)$$

## 4.2. Uncertainty Quantification via Confidence Interval

To quantify uncertainty in $\hat{\theta}$, we derive a confidence interval for $\theta$ that incorporates variance contributions from both the test and calibration datasets (Lang & Reiczigel, 2014):

$$\boxed{\begin{aligned} &\tilde{\theta} + d\tilde{\theta} \\ &\pm z_\alpha \sqrt{\frac{\frac{\tilde{p}(1-\tilde{p})}{\tilde{n}} + \left(1 - \tilde{\theta}\right)^2 \cdot \frac{\tilde{q}_0(1-\tilde{q}_0)}{\tilde{m}_0} + \tilde{\theta}^2 \cdot \frac{\tilde{q}_1(1-\tilde{q}_1)}{\tilde{m}_1}}{(\tilde{q}_0 + \tilde{q}_1 - 1)^2}}, \end{aligned}} \qquad (6)$$

where values outside the interval $[0, 1]$ are truncated to 0 or 1. Here, $z_\alpha$ denotes the $(1 - \alpha/2)$ quantile of the standard normal distribution, e.g., $z_{0.05} = 1.96$, and the adjusted quantities are defined as

$$\begin{aligned} &\tilde{n} = n + z_\alpha^2, \quad \tilde{m}_0 = m_0 + 2, \quad \tilde{m}_1 = m_1 + 2, \\ &\tilde{p} = \frac{n \cdot \hat{p} + z_\alpha^2/2}{n + z_\alpha^2}, \quad \tilde{q}_0 = \frac{m_0 \cdot \hat{q}_0 + 1}{m_0 + 2}, \\ &\tilde{q}_1 = \frac{m_1 \cdot \hat{q}_1 + 1}{m_1 + 2}, \quad \tilde{\theta} = \frac{\tilde{p} + \tilde{q}_0 - 1}{\tilde{q}_0 + \tilde{q}_1 - 1}, \end{aligned} \qquad (7)$$

$$d\tilde{\theta} = 2z_\alpha^2 \left(-(1 - \tilde{\theta}) \cdot \frac{\tilde{q}_0(1-\tilde{q}_0)}{\tilde{m}_0} + \tilde{\theta} \cdot \frac{\tilde{q}_1(1-\tilde{q}_1)}{\tilde{m}_1}\right). \qquad (8)$$

The confidence interval in (6) reflects uncertainty from both the test and calibration datasets through $\tilde{n}$, $\tilde{m}_0$, and $\tilde{m}_1$. As these sample sizes increase, the terms inside the square root decrease, resulting in a shorter confidence interval for $\theta$. Because LLM-as-a-judge evaluations can be run at scale with

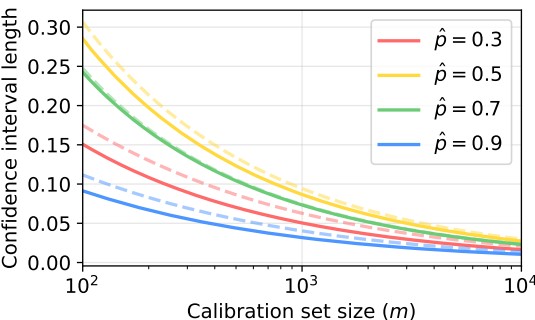

*Figure 3.* Confidence-interval length versus calibration-set size under $\hat{q}_0 = 0.7$, $\hat{q}_1 = 0.9$, and $n \to \infty$. We consider four test-set cases in which the naive estimator $\hat{p}$ takes one of the values in $\{0.3, 0.5, 0.7, 0.9\}$. Dashed curves correspond to calibration sets with symmetric allocation across label types ($m_0 = m_1$), while solid curves correspond to calibration sets using the adaptive allocation strategy in Algorithm 1, which results in shorter intervals.

minimal cost, the test-set size $n$ can often be made extremely large. In the limit $n \to \infty$, test-set uncertainty vanishes, and the interval length is determined solely by the calibration sample sizes $m_0$ and $m_1$. This observation enables practitioners to target a desired interval length and determine the minimal calibration budget required to achieve it.

Figure 3 illustrates how the confidence-interval length decreases as the calibration dataset grows. The dashed curves correspond to calibration datasets with $m_0 = m_1$. For example, when $\hat{p} = 0.3$ is estimated from the test set and $\hat{q}_0 = 0.7$, $\hat{q}_1 = 0.9$ (red dashed curve), achieving an interval shorter than 0.1 requires $m \approx 200$ calibration examples.

**Allocation Strategy to Reduce Confidence-Interval Length.** Furthermore, as the calibration dataset is collected independently, its label composition can sometimes be influenced through sampling strategies. As a result, asymmetric label sizes ($m_0 \neq m_1$) are feasible. This flexibility matters because the two label types typically contribute asymmetrically to the overall uncertainty, so an imbalanced allocation can reduce the interval length.

Motivated by the benefits of asymmetric allocation, we introduce an adaptive allocation strategy in Algorithm 1. The algorithm first collects a small pilot calibration set (e.g., $m_{\text{pilot}} = 10$ per label type) to obtain preliminary estimates $\tilde{q}_0$ and $\tilde{q}_1$. It then computes the empirical error ratio $(1 - \tilde{q}_0)/(1 - \tilde{q}_1)$ and combines it with the test-set estimate $\hat{p}$ to determine an allocation $(m_0, m_1)$ that approximately minimizes the confidence-interval length in (6). As shown by the solid curves in Figure 3, this adaptive allocation gives shorter intervals under a fixed calibration budget than symmetric allocation. The optimality of this strategy in minimizing the confidence-interval length under a fixed calibration budget is established in the following section.

# 5. Theoretical Justification of Our Method

We provide theoretical justification for the method proposed in Section 4, including derivations, bias analysis, and optimality of the allocation strategy.

## 5.1. Mitigating Bias in Point Estimator

We compare the bias-corrected estimator $\hat{\theta}$ in (5) with the naive estimator $\hat{p}$ in (2). By the law of total probability,

$$\mathbb{E}[\hat{p}] = p = (q_0 + q_1 - 1) \cdot \theta + (1 - q_0). \qquad (9)$$

Thus, $\mathbb{E}[\hat{p}] = \theta$ for all $\theta$ if and only if the LLM judge is perfect (i.e., $q_0 = q_1 = 1$); otherwise, $\hat{p}$ is biased. When $q_0 + q_1 < 2$, the expectation can be rewritten as

$$\mathbb{E}[\hat{p}] = \theta + (2 - q_0 - q_1)\left(\frac{1 - q_0}{2 - q_0 - q_1} - \theta\right),$$

which makes the direction of the bias explicit: when the true accuracy $\theta$ is smaller than the threshold $\frac{1 - q_0}{2 - q_0 - q_1}$, the estimator $\hat{p}$ exhibits a positive bias, i.e., $\mathbb{E}[\hat{p}] > \theta$; conversely, when $\theta$ exceeds this threshold, the bias becomes negative.

**Bias-Corrected Estimator.** Assuming $q_0 + q_1 > 1$, inverting the above relation in (9) gives the estimator in (4). Replacing $q_0$ and $q_1$ with their empirical estimates $\hat{q}_0$ and $\hat{q}_1$ gives the bias-corrected estimator $\hat{\theta}$ in (5).

When $q_0$ and $q_1$ are known, $\hat{\theta}$ is unbiased. When they are estimated from a calibration dataset, $\hat{\theta}$ can exhibit bias. Nevertheless, the following result shows that $\hat{\theta}$ attains smaller bias than the naive estimator $\hat{p}$ in (2) as the calibration-set size grows. All proofs are provided in Appendix E.

**Proposition 5.1.** *Suppose that $m := 2m_0 = 2m_1$ and that $q := q_0 = q_1$ with $0.5 < q \le 1$. For sufficiently large $m \gtrsim 2q/(2q - 1)^2$, the absolute bias of $\hat{\theta}$ in (1) is always smaller than that of $\hat{p}$ in (2) for all $\theta \in [0, 1]$.*

Even when $\hat{q}_0$ and $\hat{q}_1$ are estimated from data, $\hat{\theta}$ has smaller bias than $\hat{p}$ for sufficiently large calibration sets, with the required size depending on the LLM judge's sensitivity and specificity: fewer samples suffice when the LLM has high sensitivity and specificity ($q \approx 1$), while substantially more are required as these approach chance level ($q \approx 0.5$).

## 5.2. Uncertainty Quantification via Confidence Interval

We quantify uncertainty in $\hat{\theta}$ arising from two sources: the test dataset used to estimate $p$ and the calibration dataset used to estimate $q_0$ and $q_1$. Applying the delta method and the binomial variance formulas for $\hat{p}$, $\hat{q}_0$, and $\hat{q}_1$, we obtain the asymptotic variance

$$\text{Var}(\hat{\theta}) = \frac{\frac{\hat{p}(1 - \hat{p})}{n} + (1 - \hat{\theta})^2 \cdot \frac{\hat{q}_0(1 - \hat{q}_0)}{m_0} + \hat{\theta}^2 \cdot \frac{\hat{q}_1(1 - \hat{q}_1)}{m_1}}{(\hat{q}_0 + \hat{q}_1 - 1)^2}. \quad (10)$$

A detailed derivation is provided in Appendix E.1.

Based on this variance, we construct a confidence interval for $\theta$ using the *"add two successes and two failures"* adjusted Wald approach (de Laplace, 1820; Agresti & Coull, 1998; Brown et al., 2001; Lang & Reiczigel, 2014). Specifically, we replace $\hat{p}$, $\hat{q}_0$, and $\hat{q}_1$ with their adjusted versions $\tilde{p}$, $\tilde{q}_0$, and $\tilde{q}_1$, as defined in (7). These adjustments can be interpreted as adding one (or $z_\alpha^2/2$) success and one (or $z_\alpha^2/2$) failure to each estimate, improving coverage accuracy for small sample sizes (Agresti & Caffo, 2000). Substituting the estimates gives the confidence interval in (6).

The adjustment also induces a small shift in the interval center (i.e., $d\tilde{\theta}$ in (8)) due to the dependence of $\hat{\theta}$ on $\hat{q}_0$ and $\hat{q}_1$. Its effect on interval length is negligible and therefore ignored. See Lang & Reiczigel (2014) for details.

**Optimal Allocation for Minimizing Confidence-Interval Length.** We characterize the optimal allocation of calibration samples across label types under a fixed budget for minimizing the confidence-interval length. To this end, we define the error ratio $\kappa := (1 - \tilde{q}_0)/(1 - \tilde{q}_1)$.

**Proposition 5.2.** *Suppose that $\tilde{q}_0$ and $\tilde{q}_1$ are close to 1. Then the minimum length of the confidence interval defined in (6) is achieved when $\tilde{m}_0 \approx (1/\tilde{p} - 1)\sqrt{\kappa} \cdot \tilde{m}_1$.*

This result provides guidance for allocating calibration samples between the two label types. In particular, when the LLM judge is less accurate at identifying *'correct'* responses (i.e., when $\kappa$ is small) or when the naive estimator $\tilde{p}$, the proportion of test-set responses judged as *'correct'* by the LLM, is large, more calibration samples should be allocated to estimating $\tilde{q}_1$ to minimize the interval length, resulting in an optimal allocation with $m_0 < m_1$. In practice, we approximate the optimal allocation using Algorithm 1, which estimates the error ratio $\hat{\kappa}$ from a pilot calibration sample and allocates $m_0$ and $m_1$ according to Proposition 5.2.

# 6. When Are LLM-as-a-Judge Evaluations Preferable to Human-Only Evaluation?

In Section 5, we show that a human-labeled calibration dataset is essential for reliable LLM-as-a-judge evaluation because it enables bias correction. This raises a natural question: If human annotators are available, why not instead estimate the target accuracy $\theta$ directly from the human-only evaluation on the test set?

Suppose that a fixed evaluation budget of $m$ human annotations is available. One could either (a) use these $m$ annotations to construct a calibration dataset for estimating the LLM-judge's sensitivity and specificity, and then apply bias correction to LLM-based evaluations, or (b) apply them directly to $m$ test instances to estimate the true accuracy

from human evaluation alone. We compare these two strategies and show that LLM-as-a-judge evaluation with our correction method can be statistically more efficient than human-only evaluation in certain parameter regimes.

**Parameter Regimes Where LLM-as-a-Judge Achieves Lower Variance.** Human-only evaluation relies on a limited annotation budget, which restricts how many test instances can be evaluated and therefore limits how much the variance of the estimator can be reduced. In contrast, LLM-as-a-judge evaluation can be applied to an arbitrarily large test set at negligible marginal cost, so that the dominant uncertainty stems from bias correction based on a finite calibration set. These differences reflect a trade-off between a costly, near-perfect oracle that can be queried on a small number of instances, versus an inexpensive but imperfect oracle that can be deployed at scale. Importantly, as the judge becomes more reliable (i.e., as $q_0$ and $q_1$ approach one), calibration-induced uncertainty diminishes. Consequently, by aggregating large-scale LLM judgments with appropriate bias correction, it is possible to obtain an estimator whose variance is *smaller than that of human-only evaluation*.

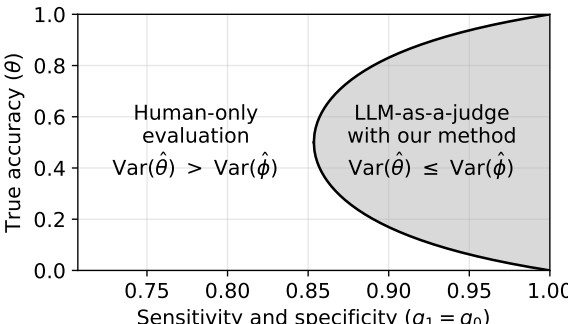

*Figure 4.* Comparison of variances between the LLM-as-a-judge estimator under our correction method, $\hat{\theta}$ in (7), and the human-only estimator, $\hat{\phi}$ in Proposition 6.1, when $q_1 = q_0$ and $m \to \infty$. Shaded regions indicate regimes where our LLM-as-a-judge evaluation is preferable to human-only evaluation in terms of variance. See Appendix D for asymmetric cases where $q_0 \neq q_1$.

Figure 4 visualizes this comparison. The shaded regions indicate parameter regimes over $q_0$, $q_1$, and $\theta$ where the bias-corrected LLM-as-a-judge estimator has smaller variance than the human-only estimator. The variance advantage is most pronounced around $\theta = 1/2$, where the intrinsic uncertainty of binary evaluation is largest, and the shaded region expands as the LLM judge becomes more accurate. The following proposition formalizes these parameter regimes.

**Proposition 6.1.** *Let $M_1 \sim \text{Binomial}(m, \theta)$ denote the number of* 'correct' *responses from human evaluation, and define the human-only estimator $\hat{\phi} := M_1/m$. Let $\hat{\theta}$ be the bias-corrected estimator in (7), where $m$ instances are used for calibration, and an infinite number of instances ($n \to \infty$) are used for testing with LLM-as-a-judge. Fix $\delta \in (0, 1)$*

*and define $\epsilon := \sqrt{\frac{\log(2/\delta)}{2m}}$. If $\epsilon < \min\{\theta, 1 - \theta\}$, then with probability at least $1 - \delta$, $\text{Var}(\hat{\theta}) \leq \text{Var}(\hat{\phi})$ whenever*

$$\theta(1-\theta) \geq \frac{1}{(q_0+q_1-1)^2}\left(\frac{(1-\theta)^2 q_0(1-q_0)}{1-\theta-\epsilon} + \frac{\theta^2 q_1(1-q_1)}{\theta-\epsilon}\right). \quad (11)$$

*Moreover, as $m \to \infty$, the sufficient condition in* (11) *becomes necessary and sufficient. If $q := q_0 = q_1$, the condition reduces to*

$$\theta \in \left[\frac{1}{2} - \sqrt{\frac{1}{2} - \frac{1}{4(2q-1)^2}}, \frac{1}{2} + \sqrt{\frac{1}{2} - \frac{1}{4(2q-1)^2}}\right].$$

It is worth clarifying why human-only evaluation can still exhibit variance even if the true label of every test instance is evaluated and therefore known. The quantity of interest is the true accuracy of the test distribution, i.e., the expected value of the true label indicating whether an instance is 'correct'. In practice, we observe a random sample of $m$ instances from the test distribution, and different samples of size $m$ result in different empirical accuracies. Thus, the variance of the estimator arises from subsampling the test distribution, not from uncertainty in individual labels.

**Positioning Contemporary LLM-as-a-Judge Approaches in the Parameter Regimes.** To contextualize Proposition 6.1, we examine where current LLM judges fall within the parameter regimes characterized above. We use the Chatbot Arena benchmark described later in Section 7.2, and estimate each judge's sensitivity $q_1$ and specificity $q_0$ by treating human preference labels as ground truth. Experimental details are provided in Appendix G.

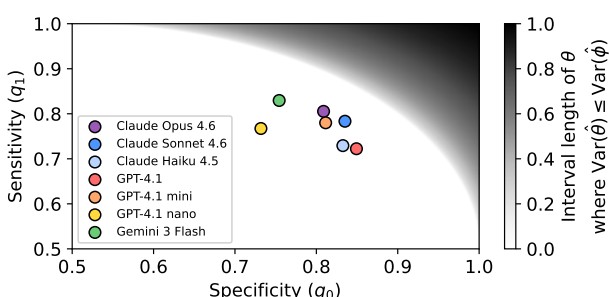

*Figure 6.* Estimated sensitivity $q_1$ and specificity $q_0$ of seven contemporary LLM-as-a-judge models on Chatbot Arena pairwise comparisons between `Alpaca-13B` and other models. Each point corresponds to one judge. The grayscale background indicates the range of $\theta$ values where calibrated LLM-as-a-judge evaluation has lower variance than human-only evaluation, according to Proposition 6.1. Darker shades indicate a wider such range.

Figure 6 maps the estimated sensitivity and specificity of each judge onto the $(q_0, q_1)$ space. The grayscale background indicates the range of $\theta$ values for which the condition in Proposition 6.1 holds. Darker shades correspond to a wider range of $\theta$ values where calibrated LLM-as-a-judge evaluation has lower variance than human-only evaluation.

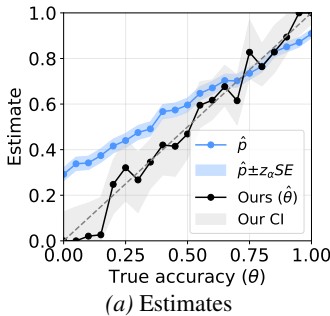
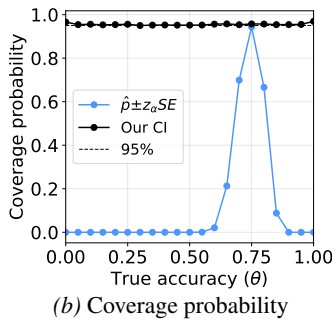
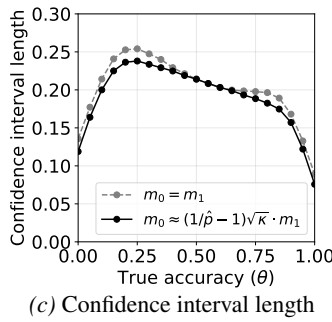

*(a)* Estimates        *(b)* Coverage probability        *(c)* Confidence interval length

*Figure 5.* Monte Carlo simulation for estimating $\theta$ under an imperfect LLM judge with $(q_0, q_1) = (0.7, 0.9)$. We evaluate estimators across 21 values of $\theta \in [0, 1]$, each visualized as a single point. Figure 5a reports the results from a single run, while Figure 5b and Figure 5c summarize averages computed over 10,000 replications. All experiments use a test dataset of size $n = 1000$ and a calibration dataset of size $m = 200$, and we use an equal allocation $m_0 = m_1$ for Figure 5a and Figure 5b. *(a)* The naive estimator $\hat{p}$ in (2) exhibits bias, while the bias-corrected estimator $\hat{\theta}$ in (5) closely recovers the true accuracy $\theta$ across all values. Shaded regions represent the 95% confidence intervals (CI). *(b)* Across all $\theta$, the coverage probability of the confidence interval remains consistently close to the nominal 95% level. *(c)* Given a fixed calibration budget of $m = 200$, we compare two allocation strategies: an equal split ($m_0 = m_1$) and the allocation proportional to $m_0 \propto (1/\hat{p} - 1)\sqrt{\kappa} \cdot m_1$ by using Algorithm 1. The proposed allocation gives shorter confidence intervals.

In this Chatbot Arena setting, none of the judges we evaluate falls inside the favorable region, so human-only evaluation still retains a variance advantage on this benchmark. However, stronger judges, such as `Claude-Opus-4.6`, lie closer to the boundary of the favorable region, suggesting that the variance advantage of calibrated LLM-based evaluation may become attainable as judge quality improves.

This conclusion depends not only on the LLM judge but also on the evaluation task. For easier tasks, the same judge can achieve higher sensitivity and specificity, and may therefore enter the favorable region. Appendix I shows a result on the AlpacaEval benchmark (Li et al., 2023), where the condition already holds when using `GPT-4.1-mini` as the judge.

## 7. Empirical Validation of Our Method

We validate the theoretical results in Section 5 through Monte Carlo simulations and real-world benchmarks.

### 7.1. Monte Carlo Simulation

We evaluate the proposed method under the following parameter configuration. The LLM judge is characterized by $(q_0, q_1) = (0.7, 0.9)$, and the true accuracy varies over $\theta \in \{0, 0.05, 0.10, \ldots, 1\}$, yielding 21 settings. This asymmetric choice of $(q_0, q_1)$ illustrates a realistic scenario where sensitivity and specificity differ. For each $(q_0, q_1, \theta)$, we generate a test dataset of size $n = 1000$ and a calibration dataset of total size $m = 200$, with equal allocation $m_0 = m_1$ unless stated otherwise.

We report the naive estimator $\hat{p}$ in (2) and its confidence interval, as well as the bias-corrected estimator $\hat{\theta}$ in (5) with the confidence interval in (6). Each configuration is replicated 10,000 times to evaluate accuracy, coverage, and interval length. Additional results across broader $(q_0, q_1)$

values are provided in Appendix J.

**Results.** Figure 5a compares $\hat{p}$ and $\hat{\theta}$ in a single simulation run. The naive estimator $\hat{p}$ is biased, particularly overestimating the true accuracy $\theta$ when $\theta$ is small. In contrast, the bias-corrected estimator $\hat{\theta}$ closely matches the true accuracy across all values of $\theta$, consistent with Proposition 5.1.

Figure 5b reports the empirical coverage probability of the confidence interval based on $\hat{p}$ and the proposed confidence interval in (6). The interval based on $\hat{p}$ attains near-zero coverage except at a few values of $\theta$. In contrast, the proposed interval achieves coverage close to the nominal 95% level across all values of $\theta$. These results confirm that our confidence interval provides reliable uncertainty quantification.

To evaluate the benefit of optimal allocation, we compare symmetric allocation ($m_0 = m_1 = 100$) with the allocation produced by Algorithm 1 under a fixed calibration budget of $m = 200$. Algorithm 1 approximately follows the optimal allocation in Proposition 5.2, and Figure 5c shows that it yields shorter confidence intervals than symmetric allocation. This benefit is consistent across broader choices of $(q_0, q_1)$ with $q_0 + q_1 > 1$, as shown in Appendix J.

### 7.2. Chatbot Arena Benchmark

We evaluate our method on the Chatbot Arena benchmark (Chiang et al., 2024), where users vote on which of two anonymous models provides the better response to the same prompt. These pairwise comparisons are aggregated into each model's win rate, defined as the probability that the target model's response is preferred over its opponent's.

We consider six target models studied in Zheng et al. (2023): `Alpaca-13B` (Taori et al., 2023), `Claude-v1`, `FastChat-T5-3B`, `GPT-4`, `LLaMA-13B` (Touvron et al., 2023), and `Vicuna-13B` (Chiang et al., 2023). For

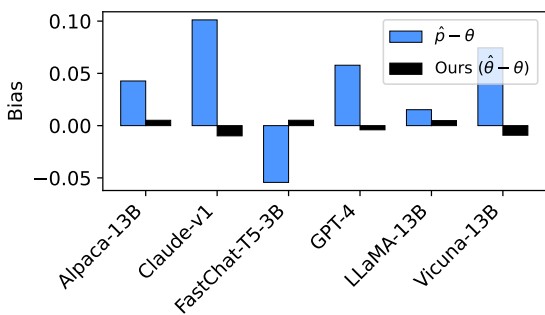

*Figure 7.* Average bias of win-rate estimates for six models on Chatbot Arena, averaged over 100 random test (90%) / calibration (10%) splits. We compare the naive LLM-based estimator $\hat{p}$ in (2) with our bias-corrected estimator $\hat{\theta}$ in (5), using `GPT-4.1-mini` as the judge. The proposed method reduces bias for all models.

*Table 1.* Empirical coverage of the 95% confidence intervals in (6) and average interval lengths for Chatbot Arena win-rate estimates.

| Model | Alpaca | Claude | FastChat-T5 | GPT-4 | LLaMA | Vicuna |
|---|---|---|---|---|---|---|
| Coverage | 96% | 95% | 97% | 97% | 99% | 99% |
| CI length | 0.193 | 0.262 | 0.256 | 0.210 | 0.267 | 0.178 |

each target model, we construct response pairs consisting of one response generated by the target model and one response generated by a randomly selected opponent model. Chatbot Arena provides human preference labels for these pairs, which we treat as ground truth for computing the target model's true win rate $\theta$. We then apply `GPT-4.1-mini` as an LLM judge to the same pairs to obtain LLM-judged preference labels, and compute the naive estimator $\hat{p}$ in (2) and our bias-corrected estimator $\hat{\theta}$ in (5).

For each trial, we randomly split the human-labeled pairs into a test set (90%) and a calibration set (10%). We estimate the judge's sensitivity and specificity on the calibration set, and evaluate the bias of $\hat{p}$ and $\hat{\theta}$ on the held-out test set. We repeat this procedure 100 times with independent random splits and report results averaged across trials.

**Results.** Figure 7 shows the average bias in win-rate estimation for each model on the Chatbot Arena benchmark. The naive estimator $\hat{p}$ exhibits non-negligible bias across all six models, whereas our estimator $\hat{\theta}$ reduces bias toward zero. Table 1 reports the coverage of the proposed 95% confidence intervals, which remains close to or slightly above the nominal level across models, indicating valid uncertainty quantification. Overall, our method provides a simple and practical bias-correction procedure for LLM-as-a-judge evaluation that requires only a small calibration dataset.

Appendix K provides additional Chatbot Arena results using `Gemini-3-Flash`, `Claude-Haiku-4.5`, and `GPT-4.1` as LLM judges, which show the same trend as in Figure 7. It also shows that raw LLM judgments can distort the ground-truth ranking of models based on Chatbot Arena win rates computed from human preference labels, while our estimator improves ranking recovery in such cases.

# 8. Robust Estimation under Distribution Shift

As mentioned in Section 1, a variety of bias-correction methods have been proposed for imperfect evaluators, typically under the assumption $\mathbb{P} = \mathbb{Q}$ between the test distribution $\mathbb{P}$ and the calibration distribution $\mathbb{Q}$. When this assumption is violated, bias correction can fail.

In practice, however, not every marginal or conditional probability under $\mathbb{P}$ need coincide with its counterpart under $\mathbb{Q}$. Instead, only certain conditional distributions of the data-generating process may remain invariant across $\mathbb{P}$ and $\mathbb{Q}$, while others may change. We therefore clarify which probabilistic assumptions are plausibly shared between $\mathbb{P}$ and $\mathbb{Q}$ in practice. Under minimal assumptions that are realistic in LLM-as-a-judge evaluation, we then characterize which methods remain unbiased under distribution shift.

**Data-Generating Process for LLM-as-a-Judge Evaluation.** LLM-as-a-judge evaluation can be formalized through a data-generating process consisting of two probabilistic components: the marginal distribution of true labels $\Pr(Z)$, which is determined by the dataset, and the conditional behavior of the LLM judge, $\Pr(\hat{Z} \mid Z, \xi)$. Here, $\xi$ denotes auxiliary factors that may influence judgments, such as response length or stylistic attributes (Zhou et al., 2023; Dubois et al., 2024).

In this section, we *only* assume that the judge's behavior depends on the true label and that it remains invariant across the test and calibration datasets:

$$\Pr_{\mathbb{P}}(\hat{Z} \mid Z) = \Pr_{\mathbb{Q}}(\hat{Z} \mid Z).$$

This assumption is motivated by common practice in LLM-as-a-judge evaluation, where the same LLM judge and prompting strategy are used for both the test and calibration datasets. In Appendix A, we discuss how this assumption can be relaxed to allow the judge behavior to depend on additional covariates $\xi$, namely $\Pr_{\mathbb{P}}(\hat{Z} \mid Z, \xi) = \Pr_{\mathbb{Q}}(\hat{Z} \mid Z, \xi)$.

In contrast, we consider the true label distributions to differ:

$$\Pr_{\mathbb{P}}(Z) \neq \Pr_{\mathbb{Q}}(Z). \tag{12}$$

Such distribution shift can arise because datasets are collected in different ways: calibration datasets are constructed to analyze how the judge makes errors, whereas test datasets reflect realistic evaluation scenarios (Jung et al., 2024). Since calibration datasets require costly human annotation, they are collected in advance and remain fixed over time. By contrast, test data are often collected online or arrive in multiple batches, each potentially having a different underlying true accuracy. Consequently, the true label distribution of the test data can differ from that of the calibration data.

Under this setup, Bayes' rule implies

$$\Pr(Z \mid \hat{Z}) \propto \Pr(\hat{Z} \mid Z)\Pr(Z),$$

*Table 2.* Existing bias-correction estimators for $\mathbb{E}_\mathbb{P}[\mathbf{Z}]$. Here, $\mathbb{P}$ denotes the test distribution and $\mathbb{Q}$ the calibration distribution. Estimators (ii)–(iv) rely on the assumption that $\Pr_\mathbb{P}(\mathbf{Z}) = \Pr_\mathbb{Q}(\mathbf{Z})$, whereas our estimator in (v) remains valid when $\Pr_\mathbb{P}(\mathbf{Z}) \neq \Pr_\mathbb{Q}(\mathbf{Z})$.

| Name | Estimation formula | Assumption |
|---|---|---|
| (i) Naive estimator ($\hat{p}$ in (2)) | $\mathbb{E}_\mathbb{P}[\hat{\mathbf{Z}}]$ | $\Pr_\mathbb{P}(\mathbf{Z}) = \Pr_\mathbb{P}(\hat{\mathbf{Z}})$ |
| (ii) Calibration-only estimator | $\mathbb{E}_\mathbb{Q}[\mathbf{Z}]$ | $\Pr_\mathbb{P}(\mathbf{Z}) = \Pr_\mathbb{Q}(\mathbf{Z})$ |
| (iii) Difference estimator (used in Prediction-Powered Inference) | $\mathbb{E}_\mathbb{P}[\hat{\mathbf{Z}}] + \mathbb{E}_\mathbb{Q}[\mathbf{Z} - \hat{\mathbf{Z}}]$ | $\Pr_\mathbb{P}(\mathbf{Z} - \hat{\mathbf{Z}}) = \Pr_\mathbb{Q}(\mathbf{Z} - \hat{\mathbf{Z}})$ |
| (iv) Conditional calibration estimator | $\Pr_\mathbb{Q}(\mathbf{Z} \mid \hat{\mathbf{Z}})\,\mathbb{E}_\mathbb{P}[\hat{\mathbf{Z}}]$ | $\Pr_\mathbb{P}(\mathbf{Z} \mid \hat{\mathbf{Z}}) = \Pr_\mathbb{Q}(\mathbf{Z} \mid \hat{\mathbf{Z}})$ |
| (v) Misclassification-adjusted estimator (Ours; $\hat{\theta}$ in (5)) | $\left(\Pr_\mathbb{Q}(\hat{\mathbf{Z}} \mid \mathbf{Z})\right)^{-1}\mathbb{E}_\mathbb{P}[\hat{\mathbf{Z}}]$ | $\Pr_\mathbb{P}(\hat{\mathbf{Z}} \mid \mathbf{Z}) = \Pr_\mathbb{Q}(\hat{\mathbf{Z}} \mid \mathbf{Z})$ |

showing that the posterior distribution depends on $\Pr(Z)$. Consequently, a shift in $\Pr(Z)$ generally induces a shift in the posterior, i.e., $\Pr_\mathbb{P}(Z \mid \hat{Z}) \neq \Pr_\mathbb{Q}(Z \mid \hat{Z})$. Estimators that assume invariance of $\Pr(Z \mid \hat{Z})$ are therefore sensitive to the distribution shift described in (12).

**Implicit Assumptions of Existing Estimators.** Following prior work (Kloos et al., 2021; Meertens et al., 2022), we compare five point estimators for estimating the true accuracy $\theta$ in (1) under the test distribution, that is, $\mathbb{E}_\mathbb{P}[Z]$. To facilitate comparison, we define

$$\mathbf{Z} := (Z, 1 - Z)^\top, \qquad \hat{\mathbf{Z}} := (\hat{Z}, 1 - \hat{Z})^\top,$$

and let $\Pr(\hat{\mathbf{Z}} \mid \mathbf{Z})$ and $\Pr(\mathbf{Z} \mid \hat{\mathbf{Z}})$ denote the confusion and calibration matrices, respectively. For example, the $(a, b)$-th entry of the confusion matrix is $[\Pr(\hat{\mathbf{Z}} \mid \mathbf{Z})]_{a,b} := \Pr([\hat{\mathbf{Z}}]_a \mid [\mathbf{Z}]_b)$, where $[\hat{\mathbf{Z}}]_a$ denotes the $a$-th component of $\hat{\mathbf{Z}}$.

Under this notation, the problem can be reformulated as estimating $\mathbb{E}_\mathbb{P}[\mathbf{Z}]$. Table 2 summarizes estimators considered in prior work. The estimator in (i) is equivalent to $\hat{p}$ in (2), since $\mathbb{E}_\mathbb{P}[\hat{\mathbf{Z}}] = (\hat{p},\, 1 - \hat{p})^\top$, and it does not use the calibration distribution $\mathbb{Q}$. In contrast, the estimators in (ii)–(v) use calibration data and rely on assumptions that certain probabilities are equal between $\mathbb{P}$ and $\mathbb{Q}$. Moreover, the estimator in (iii) coincides with the estimator used in Prediction-Powered Inference (Angelopoulos et al., 2023a). Lastly, the estimator in (v) corresponds to our estimator $\hat{\theta}$ in (5), and its equivalence is shown in Appendix E.3.

**Effect of Distribution Shift on Estimator Bias.** Most existing estimators rely on invariance assumptions that fail under the distribution shift in true accuracy described in (12), even when the judge's conditional behavior remains unchanged. In particular, the estimators in (ii)–(iv) require invariance of the marginal distribution $\Pr(Z)$, which is violated when $\Pr_\mathbb{P}(Z) \neq \Pr_\mathbb{Q}(Z)$. In contrast, the misclassification-adjusted estimator in (v) depends only on the conditional distribution $\Pr(\hat{Z} \mid Z)$ and does not rely on the marginal distribution $\Pr(Z)$.

Following the Monte Carlo simulation in Section 7, we examine estimator behavior under distribution shift. We consider a setting where the test and calibration datasets

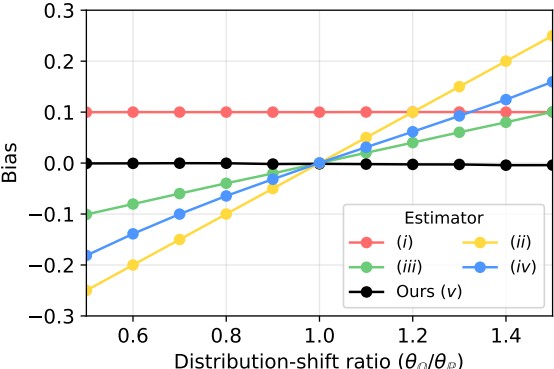

*Figure 8.* Effects of distribution shift $\Pr_\mathbb{P}(Z) \neq \Pr_\mathbb{Q}(Z)$ on estimator bias. We fix the true accuracy of the test distribution at $\theta_\mathbb{P} = 0.5$ and vary the true accuracy of the calibration distribution as $\theta_\mathbb{Q} \in [0.25, 0.75]$, corresponding to a distribution-shift ratio from 0.5 to 1.5. The misclassification-adjusted estimator in (v), used in our method, remains unbiased under this shift.

share the same conditional judge behavior, while the calibration accuracy varies as $\theta_\mathbb{Q} \in [0.25, 0.75]$, with the test accuracy held fixed at $\theta_\mathbb{P} = 0.5$.

Figure 8 reports the resulting biases of each estimator. As established in previous sections, the naive estimator in (i) is biased in this setting. Estimators (ii)–(iv) also exhibit bias when $\theta_\mathbb{P} \neq \theta_\mathbb{Q}$, whereas the misclassification-adjusted estimator in (v), used in our method, remains unbiased. Additional simulation details are provided in Appendix H.

## 9. Conclusion

In LLM-as-a-judge evaluation, noisy judgments can induce bias in naive estimates. We thus introduce a bias-corrected estimator and provide statistically sound confidence intervals. To further reduce interval length, we show that calibration design using the proposed algorithm, which allocates samples across response types, can be an effective strategy.

Beyond bias correction, we show that when the LLM judge is sufficiently accurate, our framework can achieve lower variance than human-only evaluation. Moreover, under a mild and realistic assumption on the LLM judge, our estimator remains unbiased and robust to distribution shift between the test and calibration datasets. Several directions for future work are discussed in Appendix A. We hope this work contributes to more reliable, interpretable, and transparent practices for LLM-based evaluation.

## Acknowledgements

This work was supported by the National Science Foundation (NSF) Award DMS-2023239, the NSF CAREER Award CCF-2339978, an Amazon Research Award, and a grant from FuriosaAI. In addition, it was supported by the National Research Foundation of Korea (NRF) grant funded by the Korean Ministry of Science and ICT (MSIT) (RS-2024-00345351, RS-2024-00408003), by the Institute of Information & Communications Technology Planning & Evaluation (IITP) grant funded by MSIT (RS-2023-00259934, RS-2025-02283048, RS-2026-25544647), and by the Brain Korea 21 program funded by the Korean Ministry of Education and the NRF (BK21 FOUR, No. 5199990913980).

## Impact Statement

This paper advances the field of machine learning by improving the statistical reliability of LLM-as-a-judge evaluations. By correcting bias from imperfect LLM evaluators and providing uncertainty quantification, our work helps reduce the risk of misleading performance claims and supports more rigorous evaluation practices. The contribution is methodological and does not introduce new models that affect users but instead strengthens existing evaluation pipelines. We view the broader societal impact as positive.

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

## A. Limitations and Future Work

**Limitations.**   Our analysis has several limitations. First, we treat human evaluation as the ground truth and assume that human disagreement is absent or resolved by a predefined protocol. In settings where human preferences are inherently subjective or annotators systematically disagree, the target quantity may need to be defined in terms of a distribution over human judgments. Second, the main text focuses on binary evaluations, although Appendix B shows how the same correction principle extends to evaluations with more than two categories. Third, our unbiasedness result relies on the invariance of the judge behavior between the test and calibration distributions, namely $\Pr_{\mathbb{P}}(\hat{Z} \mid Z) = \Pr_{\mathbb{Q}}(\hat{Z} \mid Z)$. This assumption may be violated if nuisance factors such as response length, style, or prompt type affect the LLM judge differently across datasets. Finally, as shown in Section 6, the current LLM judges we evaluate do not always fall in the regime where calibrated LLM-as-a-judge evaluation has lower variance than human-only evaluation. Thus, the statistical advantage of LLM-based evaluation depends on both the quality of the judge and the difficulty of the evaluation task.

**Future Work.**   Several directions remain for future work.

**(1)** Our framework could be combined with existing methods for mitigating bias in LLM judges, such as prompt-based debiasing, judge fine-tuning, or preference optimization. These methods aim to improve the judge itself, whereas our method corrects the final evaluation score after the judge has produced its labels. Thus, the two approaches are complementary. If a debiasing method improves the judge, our framework can directly benefit through improved sensitivity and specificity. At the same time, our method provides uncertainty quantification and robustness guarantees under distribution shift, which are not typically provided by judge-level debiasing methods alone. A useful direction is therefore to study how these methods can be combined, and to quantify how much judge-level debiasing improves the calibrated estimator.

**(2)** Our method can be extended to account for auxiliary factors that influence LLM-as-a-judge evaluations beyond the true label $\mathbf{Z}$. While our current analysis assumes that the LLM evaluation $\hat{\mathbf{Z}}$ depends only on the true label $\mathbf{Z}$, in practice it may also be affected by nuisance factors $\xi$, such as response length or stylistic attributes. This extension could also provide a way to model human disagreement by treating annotator-level variation as an additional factor. More generally, one may allow both the confusion matrix $\Pr_{\mathbb{Q}}(\hat{\mathbf{Z}} \mid \mathbf{Z}, \xi)$ and the LLM evaluation $\mathbb{E}_{\mathbb{P}}[\hat{\mathbf{Z}} \mid \xi]$ to depend on such factors. For example, one may estimate separate confusion matrices across strata of $\xi$, such as short and long responses, perform bias adjustment within each group, and then aggregate across strata.

**(3)** The proposed method can be extended to evaluations with more than two categories by generalizing $\left( \Pr_{\mathbb{Q}}(\hat{\mathbf{Z}} \mid \mathbf{Z}) \right)^{-1} \mathbb{E}_{\mathbb{P}}[\hat{\mathbf{Z}}]$ in Section 8. In this case, $\mathbf{Z}$ and $\hat{\mathbf{Z}}$ can be represented as probability vectors over multiple response categories. Such an extension increases the number of calibration parameters, so additional structure on the confusion matrix $\Pr_{\mathbb{Q}}(\hat{\mathbf{Z}} \mid \mathbf{Z})$ may be required for stable estimation. We provide a simple version of this extension in Appendix B.

**(4)** It would also be useful to study whether annotated preference data can be used to improve the judge itself, for example through direct preference optimization or related methods. This direction is complementary to our focus on correcting evaluations from a frozen LLM judge. Such judge adaptation may improve sensitivity and specificity, while our framework can still correct the remaining bias in the final evaluation score.

## B. Extension to Evaluations with More Than Two Categories

We mainly focus on binary evaluations, where each response is judged as either *'correct'* or *'incorrect'*. However, some evaluation protocols use more than two categories. We therefore briefly show that the same correction principle extends naturally to such settings through a confusion matrix over multiple categories.

**Multi-category correction.**   Suppose that each response is assigned one of $k$ categories. Let $Z \in [k]$ denote the human-evaluated true label, and let $\hat{Z} \in [k]$ denote the label assigned by the LLM judge. Following the notation in Section 8, we represent these labels as one-hot vectors, $\mathbf{Z} := \mathbf{e}_Z \in \mathbb{R}^k$ and $\hat{\mathbf{Z}} := \mathbf{e}_{\hat{Z}} \in \mathbb{R}^k$, where $\mathbf{e}_a$ denotes the $a$-th basis vector.

The target quantity is the distribution of human-evaluated true labels under the test distribution $\mathbb{P}$, namely $\theta := \mathbb{E}_{\mathbb{P}}[\mathbf{Z}]$. In contrast, the naive LLM-based estimator targets the distribution of LLM judgments, $\mathbb{E}_{\mathbb{P}}[\hat{\mathbf{Z}}]$. Thus, as in the binary case, directly reporting the raw LLM-judged label distribution can be biased for the human-evaluated true label distribution.

To correct this bias, define the confusion matrix of the LLM judge as $\mathrm{Pr}_{\mathbb{Q}}(\hat{\mathbf{Z}} \mid \mathbf{Z}) \in \mathbb{R}^{k \times k}$, whose $(a, b)$-th entry is

$$\left[ \mathrm{Pr}_{\mathbb{Q}}(\hat{\mathbf{Z}} \mid \mathbf{Z}) \right]_{a,b} := \mathrm{Pr}_{\mathbb{Q}}(\hat{Z} = a \mid Z = b).$$

This matrix is the multi-category analogue of the sensitivity and specificity parameters $q_1$ and $q_0$ in (3). Under the same type of invariance assumption used in Section 8, namely $\mathrm{Pr}_{\mathbb{P}}(\hat{\mathbf{Z}} \mid \mathbf{Z}) = \mathrm{Pr}_{\mathbb{Q}}(\hat{\mathbf{Z}} \mid \mathbf{Z})$, we have $\mathbb{E}_{\mathbb{P}}[\hat{\mathbf{Z}}] = \mathrm{Pr}_{\mathbb{Q}}(\hat{\mathbf{Z}} \mid \mathbf{Z})\mathbb{E}_{\mathbb{P}}[\mathbf{Z}]$. Therefore, when $\mathrm{Pr}_{\mathbb{Q}}(\hat{\mathbf{Z}} \mid \mathbf{Z})$ is nonsingular, our adjusted estimator is

$$\hat{\theta} = \widehat{\mathrm{Pr}_{\mathbb{Q}}}(\hat{\mathbf{Z}} \mid \mathbf{Z})^{-1}\mathbb{E}_{\mathbb{P}}[\hat{\mathbf{Z}}], \tag{13}$$

where the confusion matrix is estimated from the calibration dataset by

$$\left[\widehat{\mathrm{Pr}_{\mathbb{Q}}}(\hat{\mathbf{Z}} \mid \mathbf{Z})\right]_{a,b} = \frac{\sum_{j \in [m]} \mathbf{1}\{\hat{z}_j = a, z_j = b\}}{\sum_{j \in [m]} \mathbf{1}\{z_j = b\}}.$$

When $k = 2$, this reduces to estimating $\hat{q}_1$ and $\hat{q}_0$ in the binary setting. See Appendix E.3 for more details.

**Monte Carlo simulation with review score categories.** We demonstrate this extension through a Monte Carlo simulation using category names inspired by a conference paper review scale. Specifically, we consider $k = 6$ ordered categories, Strong Reject (1), Reject (2), Weak Reject (3), Weak Accept (4), Accept (5), and Strong Accept (6). These categories are used only as labels for the simulated outcomes. We generate $n = 1000$ synthetic items, each assigned a ground truth category according to the distribution in Table 3. The simulated judge scores are drawn from a predefined confusion matrix $\mathrm{Pr}(\hat{\mathbf{Z}} \mid \mathbf{Z})$ parameterized to exhibit central tendency bias. This matrix is used only for data generation and is treated as unknown during calibration. A calibration set of $m = 200$ synthetic items with observed ground truth categories is used to estimate $\mathrm{Pr}(\hat{\mathbf{Z}} \mid \mathbf{Z})$, and the estimator adjusted for misclassification in (13) is then applied to the simulated LLM-judged category distribution. Results are averaged over 10 Monte Carlo repetitions.

*Table 3.* Estimated number of papers per recommendation score. Mean and standard error are evaluated over 10 Monte Carlo repetitions.

|  | Strong Reject | Reject | Weak Reject | Weak Accept | Accept | Strong Accept |
|---|---|---|---|---|---|---|
| Ground truth | 50 | 150 | 300 | 250 | 200 | 50 |
| Naive LLM-based estimate | 39.8 (1.9) | 127.1 (2.3) | 321.6 (3.4) | 311.7 (2.4) | 156.5 (2.1) | 43.2 (1.1) |
| Misclassification-adjusted estimate | 53.7 (3.8) | 149.9 (6.0) | 279.9 (11.8) | 264.5 (11.8) | 195.0 (7.1) | 57.0 (3.7) |

Table 3 shows that the LLM-based estimate overestimates the middle categories, especially Weak Reject and Weak Accept, and underestimates the tail categories, especially Strong Reject and Strong Accept. In contrast, the misclassification-adjusted estimate substantially reduces this central tendency bias and better recovers the ground-truth distribution across all categories.

**Practical considerations.** Extending the correction beyond binary labels requires estimating a $k \times k$ confusion matrix. Thus, the number of calibration parameters grows with $k$, and stable estimation requires enough calibration examples for each category. When $k$ is large or some categories are rare, one may need to impose additional structure on the confusion matrix, such as a banded structure for ordered labels, or merge rarely used categories before applying the correction.

## C. Adaptive Strategy for Allocating Calibration Samples Across Labels

We propose an adaptive strategy for allocating calibration sample sizes across the two true-label types (*'correct'* and *'incorrect'*) using a small pilot calibration set. Leveraging pilot estimates of sensitivity $\tilde{q}_1$ and specificity $\tilde{q}_0$, Algorithm 1 attains an optimal allocation under a fixed total calibration budget, leading to shorter confidence intervals.

---

**Algorithm 1** Adaptive strategy for allocating calibration sample sizes across true label types

---

**Input:** Total calibration budget $m$, pilot sample size $2m_{\text{pilot}}$ with $2m_{\text{pilot}} \leq m$, and the estimate $\hat{p}$ from the test dataset.
**Output:** Allocated calibration sample sizes $(m_0, m_1)$.

1: **Pilot calibration.**
2:     Collect $m_{\text{pilot}}$ calibration examples with true label $z_j = 0$, and $m_{\text{pilot}}$ examples with $z_j = 1$.
3:     Compute $\tilde{q}_0$ and $\tilde{q}_1$:
$$\tilde{q}_0 = \frac{\sum_{z_j=0} \mathbf{1}\{\hat{z}_j=0, z_j=0\}+1}{m_{\text{pilot}}+2}, \quad \tilde{q}_1 = \frac{\sum_{z_j=1} \mathbf{1}\{\hat{z}_j=1, z_j=1\}+1}{m_{\text{pilot}}+2}.$$
4:     Compute the estimated error ratio: $\hat{\kappa} = \frac{1-\tilde{q}_0}{1-\tilde{q}_1}$.
5: **Compute adaptive allocation.**
6:     Using the approximation in Proposition 5.2, compute the provisional allocation:
$$m_1^{\star} = \text{round}\left(\frac{m}{1+(1/\hat{p}-1)\sqrt{\hat{\kappa}}}\right).$$
7:     Enforce pilot size: $m_1 = \min\left\{\max\{m_1^{\star}, m_{\text{pilot}}\}, m - m_{\text{pilot}}\right\}, m_0 = m - m_1$.

---

## D. Additional Parameter Regime Visualizations

We provide additional visualizations of the parameter regimes where the bias-corrected LLM-as-a-judge estimator has lower variance than the human-only estimator, as characterized in Proposition 6.1. While Figure 4 focuses on the symmetric case $q_0 = q_1$, Figure 9 shows complementary slices for asymmetric settings where $q_0 \neq q_1$.

The top row fixes the specificity $q_0$ at values in $\{0.7, 0.8, 0.9, 1.0\}$ and varies the sensitivity $q_1$ and the true accuracy $\theta$. The bottom row fixes the sensitivity $q_1$ at values in $\{0.7, 0.8, 0.9, 1.0\}$ and varies the specificity $q_0$ and $\theta$. The shaded regions indicate parameter values where the bias-corrected LLM-as-a-judge estimator has lower variance than the human-only estimator. These plots show that the favorable region expands as either sensitivity or specificity improves.

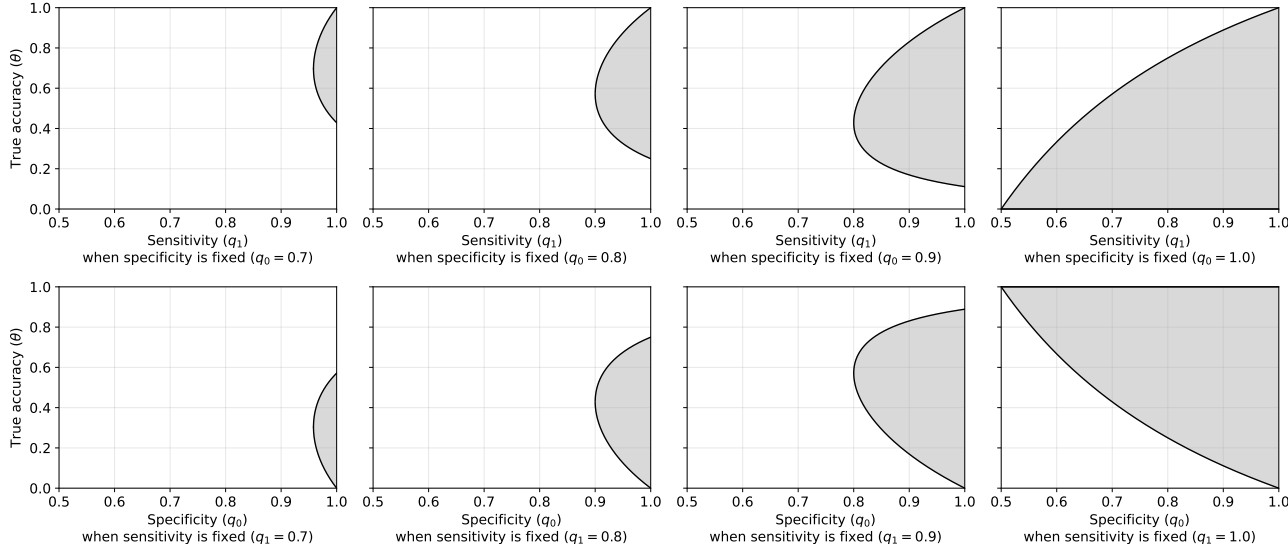

*Figure 9.* Additional comparison of variances between the LLM-as-a-judge estimator under our correction method and the human-only estimator when $q_0 \neq q_1$. The top row fixes specificity $q_0$, while the bottom row fixes sensitivity $q_1$. Shaded regions indicate regimes where our LLM-as-a-judge evaluation is preferable to human-only evaluation in terms of variance.

# E. Proofs

## E.1. Deriving the Variance of Estimators

Because $p$ follows a binomial distribution, the variance of $\hat{p}$ is

$$\mathrm{Var}(\hat{p}) = \hat{p}(1 - \hat{p})/n.$$

Similarly, we have $\mathrm{Var}(\hat{q}_0) = \hat{q}_0(1 - \hat{q}_0)/m_0$ and $\mathrm{Var}(\hat{q}_1) = \hat{q}_1(1 - \hat{q}_1)/m_1$.

We now derive the asymptotic variance of $\hat{\theta}$ using the delta method (Dorfman, 1938; Ver Hoef, 2012) for $\hat{\theta} = \frac{\hat{p} + \hat{q}_0 - 1}{\hat{q}_0 + \hat{q}_1 - 1}$. The first order derivatives with respect to $\hat{p}$, $\hat{q}_0$, and $\hat{q}_1$ are

$$\frac{\partial \hat{\theta}}{\partial \hat{p}} = \frac{1}{\hat{q}_0 + \hat{q}_1 - 1}, \qquad \frac{\partial \hat{\theta}}{\partial \hat{q}_0} = \frac{1 - \hat{\theta}}{\hat{q}_0 + \hat{q}_1 - 1}, \qquad \frac{\partial \hat{\theta}}{\partial \hat{q}_1} = \frac{-\hat{\theta}}{\hat{q}_0 + \hat{q}_1 - 1}.$$

Assuming independence between the test dataset and the calibration dataset, the delta method gives

$$\mathrm{Var}(\hat{\theta}) = \frac{\hat{p}(1 - \hat{p})/n + (1 - \hat{\theta})^2 \cdot \hat{q}_0(1 - \hat{q}_0)/m_0 + \hat{\theta}^2 \cdot \hat{q}_1(1 - \hat{q}_1)/m_1}{(\hat{q}_0 + \hat{q}_1 - 1)^2}.$$

## E.2. Proofs of Propositions

**Proposition E.1.** *Suppose that $m := 2m_0 = 2m_1$ and that $q := q_0 = q_1$ with $0.5 < q \leq 1$. For sufficiently large $m \gtrsim 2q/(2q - 1)^2$, the absolute bias of $\hat{\theta}$ in (1) is always smaller than that of $\hat{p}$ in (2) for all $\theta \in [0, 1]$.*

*Proof.* First, note that the bias of $\hat{p}$ in (2) is

$$\mathbb{E}[\hat{p}] - \theta = (q_0 + q_1 - 1)\theta + (1 - q_0) - \theta = (2\theta - 1)(1 - q).$$

Next, consider the bias of $\hat{\theta}$ in (1). By the second-order delta method, we have

$$\mathbb{E}[\hat{\theta}] \approx \frac{p + q_0 - 1}{q_0 + q_1 - 1} + \frac{1}{2}\left(-\frac{2(q_1 - p)}{(q_0 + q_1 - 1)^3} \cdot \frac{q_0(1 - q_0)}{m_0} + \frac{2(p + q_0 - 1)}{(q_0 + q_1 - 1)^3} \cdot \frac{q_1(1 - q_1)}{m_1}\right)$$

$$= \theta - \frac{(q_1 - p)}{(q_0 + q_1 - 1)^3} \cdot \frac{q_0(1 - q_0)}{m_0} + \frac{(p + q_0 - 1)}{(q_0 + q_1 - 1)^3} \cdot \frac{q_1(1 - q_1)}{m_1},$$

which implies

$$\mathbb{E}[\hat{\theta}] - \theta \approx \frac{-(1 - \theta)q_0(1 - q_0)/m_0 + \theta q_1(1 - q_1)/m_1}{(q_0 + q_1 - 1)^2} = \frac{1}{m} \cdot \frac{2q}{(2q - 1)^2} \cdot (2\theta - 1)(1 - q).$$

Hence, for sufficiently large $m$ satisfying $m \gtrsim 2q/(2q - 1)^2$, we conclude the following for all $\theta \in [0, 1]$:

$$\left|\mathbb{E}[\hat{\theta}] - \theta\right| \approx \left|\frac{1}{m} \cdot \frac{2q}{(2q - 1)^2}\right| \cdot |(2\theta - 1)(1 - q)| < |(2\theta - 1)(1 - q)| = \left|\mathbb{E}[\hat{p}] - \theta\right|.$$

$\square$

**Proposition E.2.** *Let $M_1 \sim \mathrm{Binomial}(m, \theta)$ denote the number of 'correct' responses from human evaluation, and define the human-only estimator $\hat{\phi} := M_1/m$. Let $\hat{\theta}$ be the bias-corrected estimator in (7), where $m$ instances are used for calibration, and an infinite number of instances ($n \to \infty$) are used for testing with LLM-as-a-judge. Fix $\delta \in (0, 1)$ and define $\epsilon := \sqrt{\frac{\log(2/\delta)}{2m}}$. If $\epsilon < \min\{\theta, 1 - \theta\}$, then with probability at least $1 - \delta$,*

$$\mathrm{Var}(\hat{\theta}) \leq \mathrm{Var}(\hat{\phi}) \quad whenever \quad \theta(1 - \theta) \geq \frac{1}{(q_0 + q_1 - 1)^2}\left(\frac{(1 - \theta)^2}{1 - \theta - \epsilon} \cdot q_0(1 - q_0) + \frac{\theta^2}{\theta - \epsilon} \cdot q_1(1 - q_1)\right). \quad (14)$$

*Moreover, as $m \to \infty$ the sufficient condition in (14) reduces to the necessary and sufficient condition given by*

$$\theta \in \left[ \frac{1+\Delta}{2} - \sqrt{\frac{(1+\Delta)^2}{4} - \frac{q_0(1-q_0)}{(q_0+q_1-1)^2}}, \ \frac{1+\Delta}{2} + \sqrt{\frac{(1+\Delta)^2}{4} - \frac{q_0(1-q_0)}{(q_0+q_1-1)^2}} \right], \tag{15}$$

*where we define* $\Delta := \frac{q_0(1-q_0)-q_1(1-q_1)}{(q_0+q_1-1)^2}$. *Furthermore, if* $q := q_0 = q_1 \in \left( \frac{1}{2} + \frac{1}{2\sqrt{2}}, 1 \right)$, *this reduces to*

$$\theta \in \left[ \frac{1}{2} - \sqrt{\frac{1}{2} - \frac{1}{4(2q-1)^2}}, \ \frac{1}{2} + \sqrt{\frac{1}{2} - \frac{1}{4(2q-1)^2}} \right]. \tag{16}$$

*Proof.* Since $M_1 \sim \text{Binomial}(m, \theta)$, we have

$$\text{Var}(\hat{\phi}) = \frac{\theta(1-\theta)}{m}.$$

From (10), as $n \to \infty$, the first term in the numerator vanishes, and hence

$$\text{Var}(\hat{\theta}) = \frac{(1-\theta)^2 \cdot \frac{q_0(1-q_0)}{m-M_1} + \theta^2 \cdot \frac{q_1(1-q_1)}{M_1}}{(q_0+q_1-1)^2}. \tag{17}$$

Since $\hat{\phi} = M_1/m$, Hoeffding's inequality implies that for any $\delta \in (0,1)$,

$$\Pr\left( |\hat{\phi} - \theta| \geq \epsilon \right) \leq \delta, \qquad \epsilon := \sqrt{\frac{\log(2/\delta)}{2m}}.$$

Assume that $\epsilon > 0$ is sufficiently small such that $\epsilon < \min\{\theta, 1-\theta\}$. Then, with probability at least $1 - \delta$, we have $|\hat{\phi} - \theta| \leq \epsilon$. This implies $M_1 \geq m(\theta - \epsilon)$ and $m - M_1 \geq m(1 - \theta - \epsilon)$, and therefore

$$\frac{1}{M_1} \leq \frac{1}{m(\theta - \epsilon)}, \qquad \frac{1}{m - M_1} \leq \frac{1}{m(1 - \theta - \epsilon)}.$$

Substituting these bounds into (17) gives

$$\text{Var}(\hat{\theta}) \leq \frac{(1-\theta)^2 \cdot \frac{q_0(1-q_0)}{m(1-\theta-\epsilon)} + \theta^2 \cdot \frac{q_1(1-q_1)}{m(\theta-\epsilon)}}{(q_0+q_1-1)^2} = \frac{1}{m} \cdot \frac{1}{(q_0+q_1-1)^2} \left( \frac{(1-\theta)^2}{1-\theta-\epsilon} \cdot q_0(1-q_0) + \frac{\theta^2}{\theta-\epsilon} \cdot q_1(1-q_1) \right).$$

Therefore, with probability at least $1 - \delta$, the inequality $\text{Var}(\hat{\theta}) \leq \text{Var}(\hat{\phi})$ holds whenever

$$\theta(1-\theta) \geq \frac{1}{(q_0+q_1-1)^2} \left( \frac{(1-\theta)^2}{1-\theta-\epsilon} \cdot q_0(1-q_0) + \frac{\theta^2}{\theta-\epsilon} \cdot q_1(1-q_1) \right),$$

which proves (14).

Furthermore, as $m \to \infty$, we have $\epsilon = \sqrt{\log(2/\delta)/(2m)} \to 0$, and the condition converges to

$$\theta(1-\theta) \geq \frac{q_0(1-q_0) - \theta\big(q_0(1-q_0) - q_1(1-q_1)\big)}{(q_0+q_1-1)^2},$$

which is necessary and sufficient for $\text{Var}(\hat{\theta}) \leq \text{Var}(\hat{\phi})$.

Define $\Delta := \frac{q_0(1-q_0)-q_1(1-q_1)}{(q_0+q_1-1)^2}$. Then the inequality becomes

$$\theta^2 - \theta(1+\Delta) + \frac{q_0(1-q_0)}{(q_0+q_1-1)^2} \leq 0,$$

which gives (15). Finally, when $q := q_0 = q_1 \in \left( \frac{1}{2} + \frac{1}{2\sqrt{2}}, 1 \right)$, this further reduces to (16). $\qquad\square$

### E.3. Equivalence between the Misclassification-Adjusted Estimator and Our Estimator

We show that the misclassification-adjusted estimator $\left(\mathrm{Pr}_{\mathbb{Q}}(\hat{\mathbf{Z}} \mid \mathbf{Z})\right)^{-1} \mathbb{E}_{\mathbb{P}}[\hat{\mathbf{Z}}]$ in Section 8 is equivalent to our estimator $\hat{\theta}$ in (5). From the definition, we have

$$\left(\mathrm{Pr}_{\mathbb{Q}}(\hat{\mathbf{Z}} \mid \mathbf{Z})\right)^{-1} \mathbb{E}_{\mathbb{P}}[\hat{\mathbf{Z}}] = \begin{pmatrix} \mathrm{Pr}_{\mathbb{Q}}(\hat{Z}=1 \mid Z=1) & \mathrm{Pr}_{\mathbb{Q}}(\hat{Z}=1 \mid Z=0) \\ \mathrm{Pr}_{\mathbb{Q}}(\hat{Z}=0 \mid Z=1) & \mathrm{Pr}_{\mathbb{Q}}(\hat{Z}=0 \mid Z=0) \end{pmatrix}^{-1} \begin{pmatrix} \mathrm{Pr}_{\mathbb{P}}(\hat{Z}=1) \\ \mathrm{Pr}_{\mathbb{P}}(\hat{Z}=0) \end{pmatrix}$$

$$= \begin{pmatrix} \hat{q}_1 & 1-\hat{q}_0 \\ 1-\hat{q}_1 & \hat{q}_0 \end{pmatrix}^{-1} \begin{pmatrix} \hat{p} \\ 1-\hat{p} \end{pmatrix}$$

$$= \frac{1}{\hat{q}_0 + \hat{q}_1 - 1} \begin{pmatrix} \hat{q}_0 & -(1-\hat{q}_0) \\ -(1-\hat{q}_1) & \hat{q}_1 \end{pmatrix} \begin{pmatrix} \hat{p} \\ 1-\hat{p} \end{pmatrix},$$

provided that $\hat{q}_0 + \hat{q}_1 \neq 1$. This gives

$$\left(\mathrm{Pr}_{\mathbb{Q}}(\hat{\mathbf{Z}} \mid \mathbf{Z})\right)^{-1} \mathbb{E}_{\mathbb{P}}[\hat{\mathbf{Z}}] = \frac{1}{\hat{q}_0 + \hat{q}_1 - 1} \begin{pmatrix} \hat{p} + \hat{q}_0 - 1 \\ \hat{q}_1 - \hat{p} \end{pmatrix}.$$

In particular, the first component corresponds to the estimator of $\theta = \mathrm{Pr}_{\mathbb{P}}(Z=1)$:

$$\hat{\theta} = \frac{\hat{p} + \hat{q}_0 - 1}{\hat{q}_0 + \hat{q}_1 - 1},$$

which exactly matches $\hat{\theta}$ in (5).

## F. Code

All code used for this paper, including a plug-in Python implementation of the introduced method for LLM-as-a-judge evaluation, is available at https://github.com/UW-Madison-Lee-Lab/LLM-judge-reporting. To make this appendix self-contained, we provide below the key functions that compute the bias-corrected estimator and its confidence interval, corresponding to the method described in Section 4.

```python
from math import sqrt
from scipy.stats import norm

def clip(x, low=0.0, high=1.0):
    return max(low, min(high, x))

def point_estimator(p, q0, q1):
    """Compute the adjusted point estimate."""
    th = (p+q0-1)/(q0+q1-1)
    return clip(th)

def confidence_interval(p, q0, q1, n, m0, m1, alpha=0.05):
    """Compute the adjusted (1 - alpha) confidence interval."""
    z = norm.ppf(1-alpha/2)
    p, q0, q1 = (n*p+z**2/2)/(n+z**2), (m0*q0+1)/(m0+2), (m1*q1+1)/(m1+2)
    n, m0, m1 = n+z**2, m0+2, m1+2
    th = (p+q0-1)/(q0+q1-1)
    dth = 2*z**2*(-(1-th)*q0*(1-q0)/m0+th*q1*(1-q1)/m1)
    se = sqrt(p*(1-p)/n+(1-th)**2*q0*(1-q0)/m0+th**2*q1*(1-q1)/m1)/(q0+q1-1)
    return clip(th+dth-z*se), clip(th+dth+z*se)
```

*Figure 10.* Python code implementation of the adjustment method described in Section 4 that computes the bias-corrected estimate and the $(1-\alpha)$ confidence interval for the true accuracy $\theta$. The inputs p, q0, and q1 are empirical estimates from the test and calibration datasets.

## G. Experimental Setup for Positioning Current LLM Judges

We provide details on how we estimate the location of current LLM judges in Figure 6. We use the Chatbot Arena pairwise comparison data described in Section 7.2. To estimate each judge's sensitivity and specificity, we treat human preference labels as ground truth and compare them with the corresponding LLM-judged preference labels.

For this analysis, we focus on pairwise comparisons involving `Alpaca-13B`. Each comparison consists of one response generated by `Alpaca-13B` and one response generated by an opposing model. We define the positive class as the event that the human preference label favors the `Alpaca-13B` response. Let $Z$ denote the human preference label and let $\hat{Z}$ denote the LLM-judged preference label. For each LLM judge, we estimate

$$\hat{q}_1 := \frac{\sum_{j \in [m]} \mathbf{1}\{\hat{z}_j = 1, z_j = 1\}}{\sum_{j \in [m]} \mathbf{1}\{z_j = 1\}}, \qquad \hat{q}_0 := \frac{\sum_{j \in [m]} \mathbf{1}\{\hat{z}_j = 0, z_j = 0\}}{\sum_{j \in [m]} \mathbf{1}\{z_j = 0\}}.$$

Thus, $\hat{q}_1$ measures how often the judge agrees with humans when humans prefer `Alpaca-13B`, while $\hat{q}_0$ measures how often the judge agrees with humans when humans prefer the opposing model.

We evaluate `Claude-Opus-4.6`, `Claude-Sonnet-4.6`, `Claude-Haiku-4.5`, `GPT-4.1`, `GPT-4.1-mini`, `GPT-4.1-nano`, and `Gemini-3-Flash`, and plot their estimated $(\hat{q}_0, \hat{q}_1)$ values in Figure 6. The grayscale background is computed from the condition in Proposition 6.1. For each pair $(q_0, q_1)$, we identify the set of $\theta$ values for which the calibrated LLM-as-a-judge estimator has lower variance than the human-only estimator. Darker regions correspond to a wider range of such $\theta$ values.

## H. Experimental Setup for Distribution-Shift Analysis

We describe the experimental setup used to produce the distribution-shift results in Figure 8. The setup matches the Monte Carlo simulation in Section 7, except that we vary the true accuracy of the calibration dataset.

We fix the LLM-judge behavior to $(q_0, q_1) = (0.7, 0.9)$ and the test-set accuracy to $\theta_\mathbb{P} = 0.5$, while varying the calibration-set accuracy over $\theta_\mathbb{Q} \in [0.25, 0.75]$. For each setting $(\theta_\mathbb{P}, \theta_\mathbb{Q})$, we generate a test dataset of size 1000 with $\Pr_\mathbb{P}(Z = 1) = \theta_\mathbb{P}$ and a calibration dataset of size 200 with $\Pr_\mathbb{Q}(Z = 1) = \theta_\mathbb{Q}$. We average results over 10,000 Monte Carlo replications.

## I. Additional Results on AlpacaEval Benchmark

We provide additional results on the AlpacaEval benchmark (Li et al., 2023), focusing on win rate estimation for `GPT-4`. This experiment complements the Chatbot Arena analysis in Figure 6 by showing that the condition in Proposition 6.1 depends not only on the LLM judge but also on the evaluation task.

As in Appendix G, we treat human preference labels as ground truth and estimate each judge's sensitivity $q_1$ and specificity $q_0$ by comparing LLM-judged preference labels with human preference labels. We evaluate `Claude-Sonnet-4.6`, `Claude-Haiku-4.5`, `GPT-4.1-mini`, and `GPT-4.1-nano` as LLM judges.

**Results.** Figure 11 maps the estimated sensitivity and specificity of each judge onto the $(q_0, q_1)$ space. Unlike the Chatbot Arena setting in Figure 6, `GPT-4.1-mini` falls inside the favorable region. This shows that the variance advantage of calibrated LLM-as-a-judge evaluation is not fundamentally unattainable in practice. Rather, whether the condition holds depends on both the quality of the judge and the difficulty of the evaluation task.

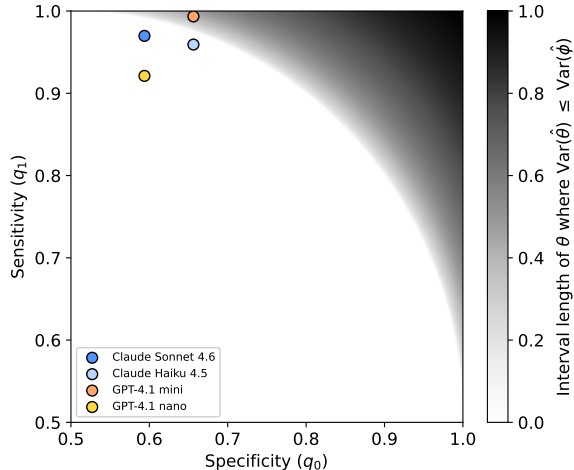

*Figure 11.* Estimated sensitivity $q_1$ and specificity $q_0$ of four LLM judges on AlpacaEval when estimating the win rate of `GPT-4`. Each point corresponds to one judge. The grayscale background indicates the range of $\theta$ values for which the variance condition in Proposition 6.1 holds, with darker shades indicating a wider favorable range.

# J. Additional Results on Monte Carlo Simulation

To complement the main simulation results in Figure 5, we report additional Monte Carlo experiments across multiple configurations with test set size $n = 1000$, calibration sizes $m \in \{200, 500\}$, and $(q_0, q_1) \in \{(0.9, 0.9), (0.7, 0.7), (0.9, 0.7), (0.7, 0.9)\}$. All other aspects of the simulation design follow the main-text setup.

Across all $(n, m, q_0, q_1)$ configurations, the main simulation findings remain consistent: bias correction improves estimation accuracy, empirical coverage matches the nominal level, and optimized calibration allocation produces shorter confidence intervals. Each figure corresponds to a specific $(n, m, q_0, q_1)$ setting and contains three subplots.

## J.1. Results for $n = 1000$ and $m = 200$

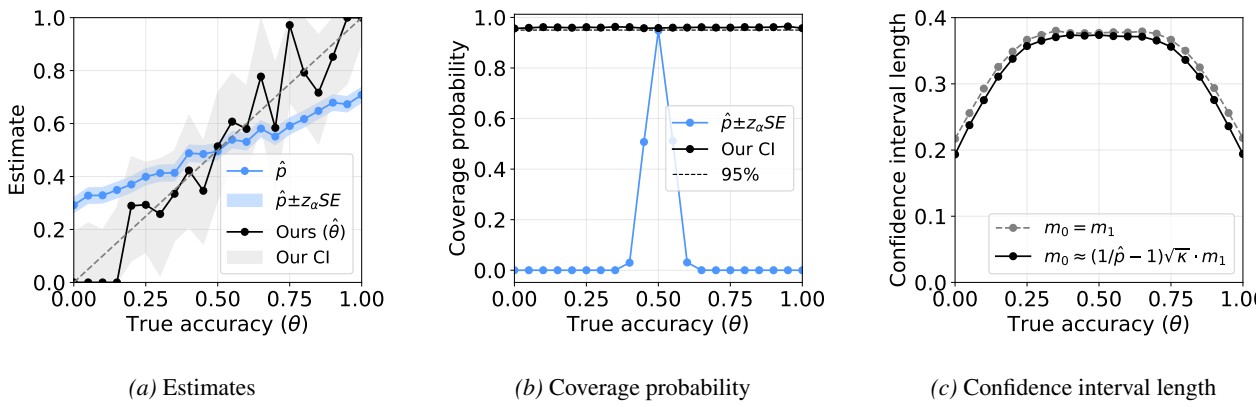

*(a)* Estimates       *(b)* Coverage probability       *(c)* Confidence interval length

*Figure 12.* Monte Carlo results for $(n, m, q_0, q_1) = (1000, 200, 0.7, 0.7)$.

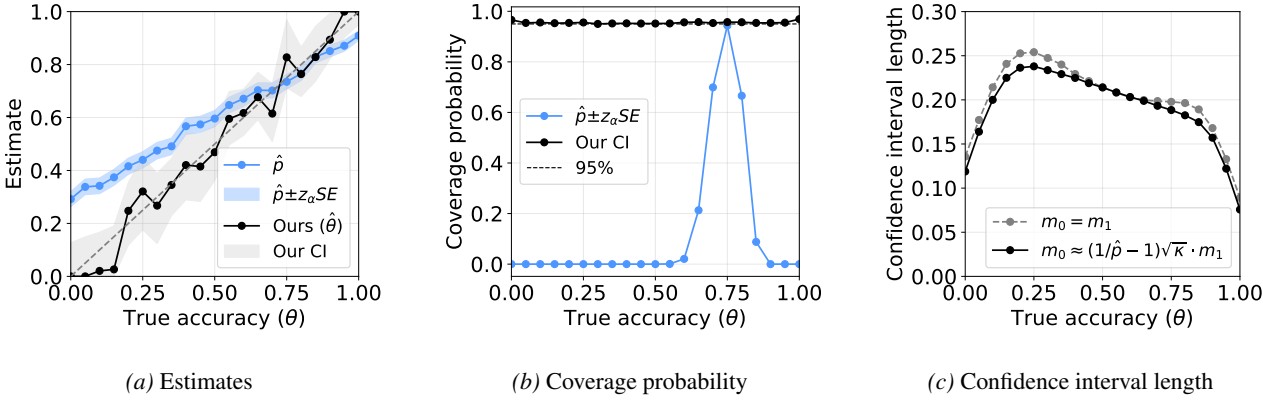

*(a)* Estimates       *(b)* Coverage probability       *(c)* Confidence interval length

*Figure 13.* Monte Carlo results for $(n, m, q_0, q_1) = (1000, 200, 0.7, 0.9)$.

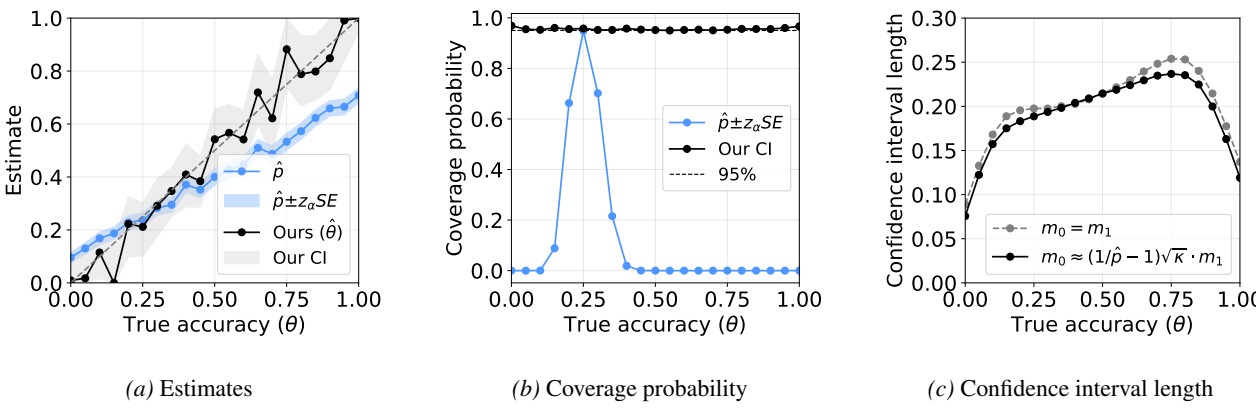

*(a)* Estimates      *(b)* Coverage probability      *(c)* Confidence interval length

*Figure 14.* Monte Carlo results for $(n, m, q_0, q_1) = (1000, 200, 0.9, 0.7)$.

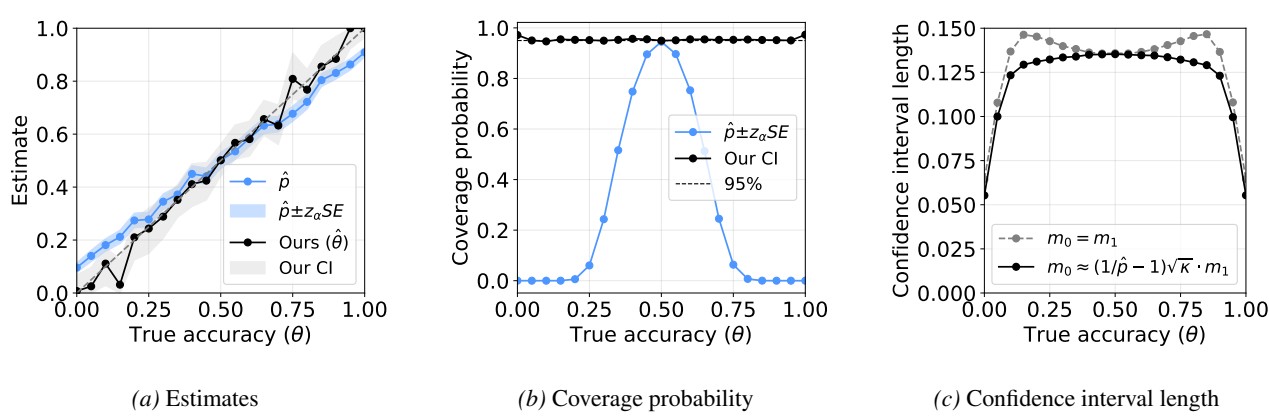

*(a)* Estimates      *(b)* Coverage probability      *(c)* Confidence interval length

*Figure 15.* Monte Carlo results for $(n, m, q_0, q_1) = (1000, 200, 0.9, 0.9)$.

## J.2. Results for $n = 1000$ and $m = 500$

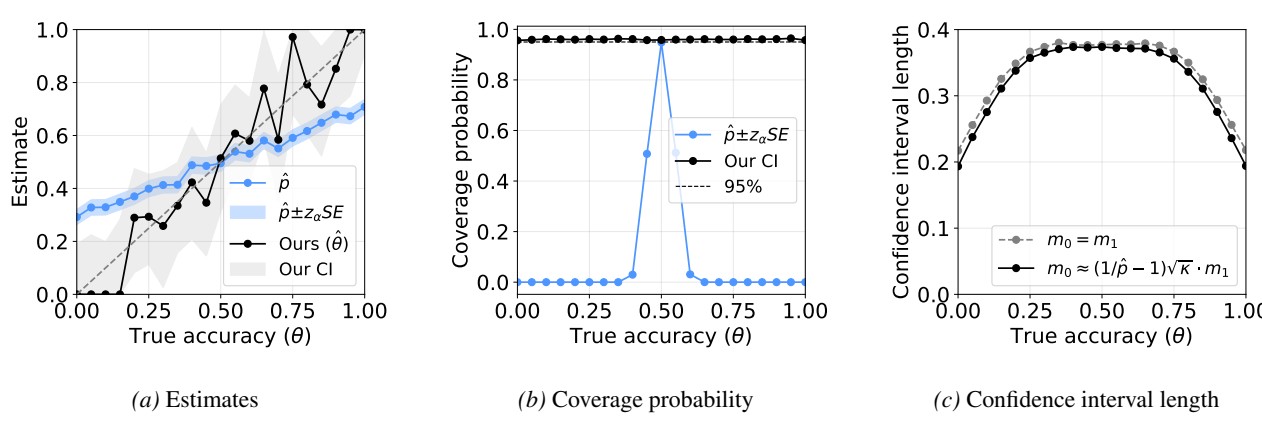

*(a)* Estimates      *(b)* Coverage probability      *(c)* Confidence interval length

*Figure 16.* Monte Carlo results for $(n, m, q_0, q_1) = (1000, 500, 0.7, 0.7)$.

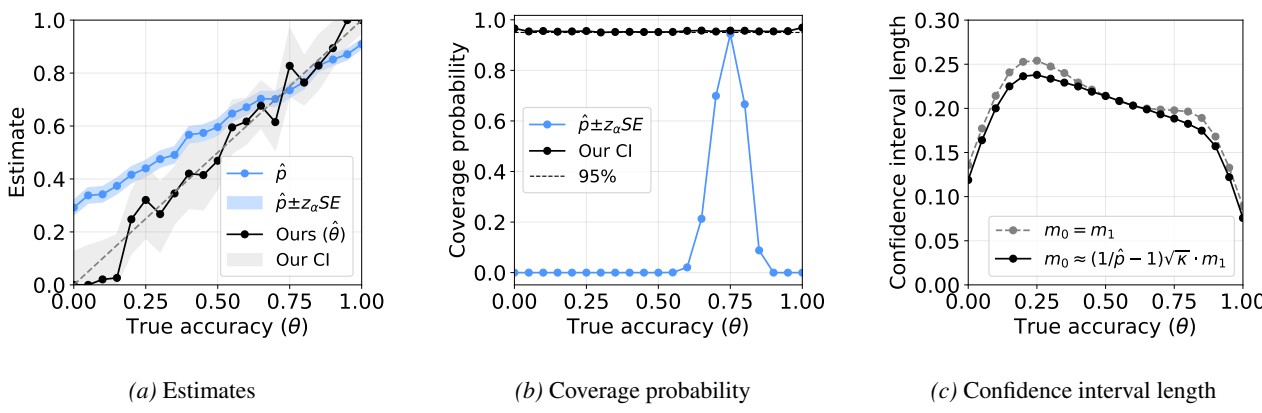

*(a)* Estimates          *(b)* Coverage probability          *(c)* Confidence interval length

*Figure 17.* Monte Carlo results for $(n, m, q_0, q_1) = (1000, 500, 0.7, 0.9)$.

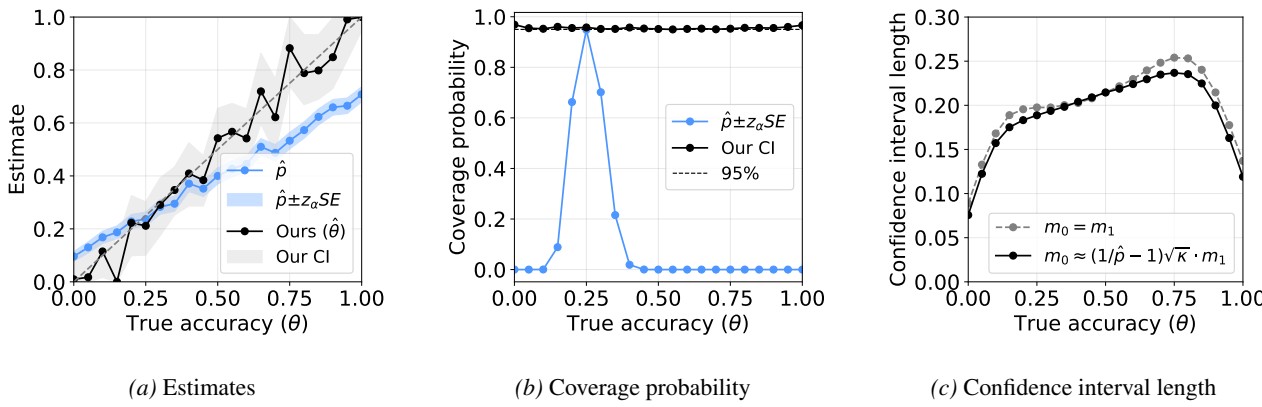

*(a)* Estimates          *(b)* Coverage probability          *(c)* Confidence interval length

*Figure 18.* Monte Carlo results for $(n, m, q_0, q_1) = (1000, 500, 0.9, 0.7)$.

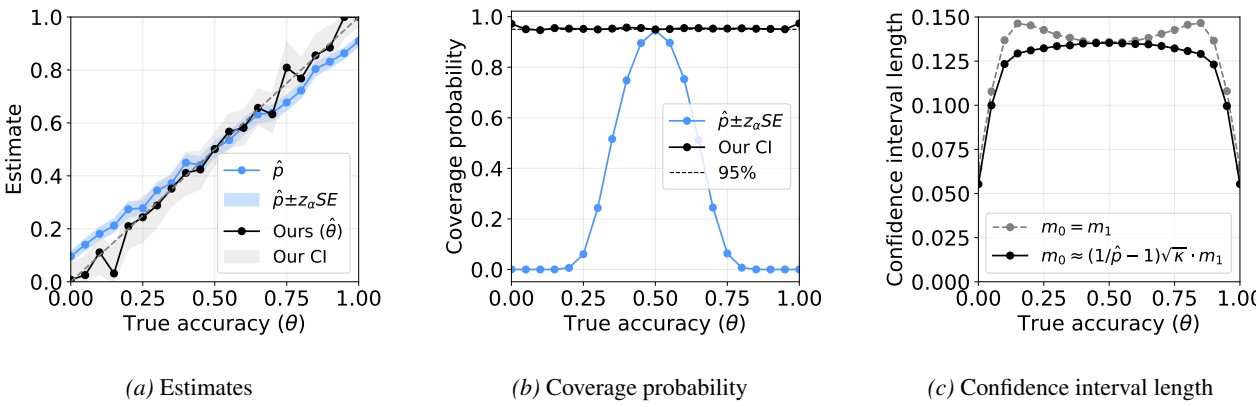

*(a)* Estimates          *(b)* Coverage probability          *(c)* Confidence interval length

*Figure 19.* Monte Carlo results for $(n, m, q_0, q_1) = (1000, 500, 0.9, 0.9)$.

## K. Additional Results on Chatbot Arena Benchmark

We extend the Chatbot Arena experiment in Section 7.2 to three additional LLM judges: GPT-4.1, Gemini-3-Flash, and Claude-Haiku-4.5. All other experimental settings are identical to those in Section 7.2. For each judge, we obtain LLM-judged labels on the same response pairs and compute the naive estimator $\hat{p}$ in (2) and our estimator $\hat{\theta}$ in (5).

**Bias in win-rate estimation.** Figure 20 shows the average bias in win-rate estimation for each model on the Chatbot Arena benchmark. Across all three additional judges, the naive estimator $\hat{p}$ exhibits bias across the six models, whereas our estimator $\hat{\theta}$ reduces bias toward zero. These results are consistent with the GPT-4.1-mini results in Figure 7, showing that the proposed method reduces bias across different judge models.

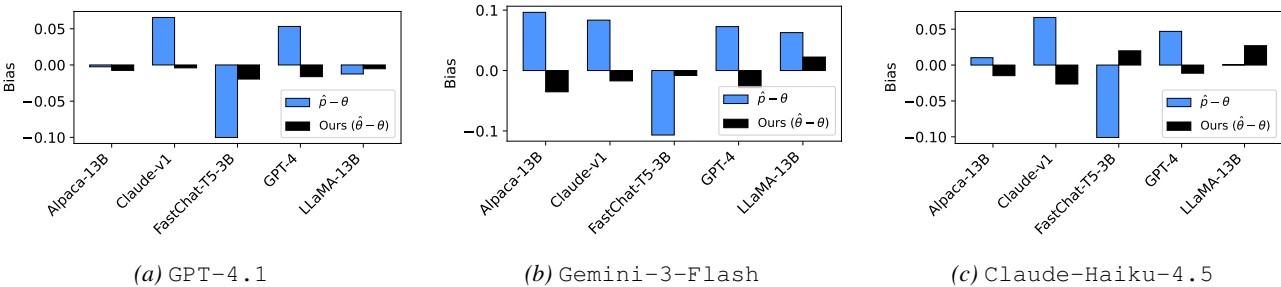

*(a)* GPT-4.1      *(b)* Gemini-3-Flash      *(c)* Claude-Haiku-4.5

*Figure 20.* Average bias of win-rate estimates for five models on Chatbot Arena, averaged over 100 random test (90%) / calibration (10%) splits. We compare the naive LLM-based estimator $\hat{p}$ in (2) with our estimator $\hat{\theta}$ in (5), using three additional LLM judges.

**Ranking recovery.** We further evaluate whether the bias in win rate estimation affects model ranking. We first compute the ground-truth ranking from the human preference labels. The resulting ranking is GPT-4 (0.850), Claude-v1 (0.786), Vicuna-13B (0.621), Alpaca-13B (0.358), FastChat-T5-3B (0.300), and LLaMA-13B (0.209). For each random split, we rank the six target models by either the naive estimator $\hat{p}$ or our estimator $\hat{\theta}$, and evaluate whether the estimated ranking matches this ground truth ranking. We report Kendall's $\tau$ and the percentage of cases where the full ranking is exactly recovered. Unless stated otherwise, we use the same 90% test and 10% calibration split as in Section 7.2. For GPT-4.1, we additionally report results with a larger calibration set, using a 60% test and 40% calibration split.

*Table 4.* Ranking recovery on Chatbot Arena over 100 random test and calibration splits. Exact ranking recovery denotes the percentage of splits in which the estimated ranking exactly matches the ground-truth ranking computed from human preference labels.

| Judge | Estimator | Kendall's $\tau$ | Exact ranking recovery |
|---|---|---|---|
| Gemini-3-Flash | Naive estimator $\hat{p}$ | 0.867 | 0% |
| Gemini-3-Flash | Our estimator $\hat{\theta}$ | 0.891 | 37% |
| Claude-Haiku-4.5 | Naive estimator $\hat{p}$ | 0.876 | 7% |
| Claude-Haiku-4.5 | Our estimator $\hat{\theta}$ | 0.900 | 40% |
| GPT-4.1 | Naive estimator $\hat{p}$ | 0.981 | 86% |
| GPT-4.1 | Our estimator $\hat{\theta}$ | 0.915 | 46% |
| GPT-4.1 with 40% calibration | Naive estimator $\hat{p}$ | 0.948 | 61% |
| GPT-4.1 with 40% calibration | Our estimator $\hat{\theta}$ | 0.967 | 75% |

Table 4 shows that raw LLM judgments can change model rankings. For Gemini-3-Flash, the naive estimator never recovers the full ground truth ranking over 100 random splits, whereas our estimator recovers it in 37% of the splits. A similar trend appears for Claude-Haiku-4.5, where exact ranking recovery improves from 7% to 40%. For GPT-4.1, the naive ranking is already largely correct, and estimating the correction from a limited calibration set introduces additional variance. When the calibration fraction is increased to 40%, however, our estimator improves both Kendall's $\tau$ and exact ranking recovery over the naive estimator. These results show that uncorrected LLM judgments can affect not only the absolute win rate estimates but also the resulting model ranking.

