# OpenReview forum: "How to Correctly Report LLM-as-a-Judge Evaluations"
_ICML.cc/2026/Conference — ICML 2026 regular_

### Official Review · Reviewer_EwMQ · 2026-02-21

**Soundness:** 2
**Presentation:** 3
**Significance:** 4
**Originality:** 3
**Overall Recommendation:** 4
**Confidence:** 3

**Summary:**

The authors adopt a calibration-based approach originally proposed for veterinary medicine by Lang & Reiczigel (2014) to the problem of estimating accuracy for LLM-as-a-Judge scenarios. Specifically, their approach estimates specificity and sensitivity of the LLM's judgment on a human annotated sample of the LLM's judgment and then infers corrected accuracy and confidence intervals.
The authors then proceed by discussing the theory behind their method and defining parameters parameter regimes for which their method (i.e. correcting the accuracy of $N$ generated LLM judgements based on a human annotated i.i.d. sample of size $n$ from $N$) is preferable over directly annotating a sample of size $n$. They conclude that their method is generally preferable if *sensitivity* = *specificity* = $q$ of the LLM judgements are roughly .85 or higher, and the true accuracy is $\in \left[.5 - \sqrt{.5 - \frac{1}{4(2q-1)^2}},~.5 + \sqrt{.5 - \frac{1}{4(2q-1)^2}} \right]$.
Furthermore, they perform case-based empirical evaluation,  ...

 - ... simulating an LLM judge with *sensitivity $\in$  {0.7, 0.9}* and *specificity $\in$  {0.7, 0.9}*, showing that their method outperforms a naive approach and that sampling the calibration set according to their Proposition 4.2 produces smaller confidence intervals than sampling the calibration set with an equal ratio of judgement outcomes.
- ... performing pairwise comparisons of LLM output quality both by human annotators and their proposed method, and showing that for *GPT-4.1-mini*, confidence intervalls are shorter than for human annotators.

Lastly, the authors show that their approach remains unbiased under distribution shift while this is not the case for other baselines (including one approach identified in related literature).

**Compliance With Llm Reviewing Policy:**

Affirmed.

**Final Justification:**

The authors' reply was sufficiently backed up with empirical evidence to change my assessment to "weak accept". Nevertheless, the new results will require some effort in rewriting the manuscript, and I strongly employ the authors to discuss the impact of the new results for their variance analysis in the limitations and discussion parts of the paper. Since I cannot guarantee that this will happen, I can not, in  good faith, go beyond "weak accept".

**Key Questions For Authors:**

A satisfactory answer to the following questions, along with the corresponding changes to the paper, would sway my assessment of the paper to accept:

1. You present sensitivity, specificity, and accuracy regimes where your method for corrected LLM-as-a-Judge evaluation achieves lower variance than direct human evaluation in Figure 4. Did you analyse where contemporary LLM-as-a-Judge approaches are located in terms of these parameters and whether they can meet the relatively high requirements indicated by the Figure?
2. Can you provide the results of the experiment in Section 6.2 for at least one more LLM judge to indicate model dependency of your approach?
3. Is it possible for you to share the code of your experiments, too? If so, please also provide the code in your reply to this review and upload it to your GitHub repository.

**Limitations:**

If not addressed in the final paper, the following limitation should be explicitly mentioned:

"The results Section 5 are not generalisable to choices of *sensitivity* and *specificity* other than *sensitivity = specificity*."

The potential negative societal impact of the work has been adequately addressed.

**Strengths And Weaknesses:**

**Strengths:**

- The paper has high **significance** as it tackles the highly relevant problem of making LLM-as-a-Judge approaches, which are found all over NLP literature these days, more reliable. To the best of my knowledge, and based on a quick literature sweep, this topic remains underexplored as of now.
- In terms of **presentation**, the paper is overall well written and structured.
- The novelty of the paper lies in applying a method proposed for science in veterinary medicine to LLM-as-a-Judge systems. Since this application seems generally thought through and due to the high relevance of the topic, **originality** can be seen as one of the strengths of the paper.
- Generally, the technical part of the paper seems **sound**, although I did not check the proofs in the appendix.

**Weaknesses:**

- The empirical evaluation of the proposed method is largely based on case studies (*Section 6*) and simulated scenarios without real LLM involvement (*Sections 6.1 and 7*). Furthermore, their experiments are computed on only one LLM judge (*GPT-4.1-mini*) in one such case-study experiment. **For a *sound* empirical evaluation, I would expect a comparison of different LLM-judges.**
- The choice of *sensitivity = specificity* for the analysis in Section 5 does not generalise to real-world scenarios, which makes the findings of this chapter practically irrelevant. **This could, for example, be improved by including values for a human annotation scenario in Figures 9c to 16c.**
- The sensitivity and specificity parameters for the Monte Carlo Simulation (*Section 6.1*) are selected without justification, and from only reading the main body of the paper, it remains unclear how these results generalise to other choices. Nevertheless, the authors provide results for other choices of sensitivity and specificity in Appendix F without discussing them in the main text. **In order to improve generalisability, the results from Appendix F should be explicitly discussed in the main text**.
- Lastly, the authors do not provide the code for replicating their experiments (*the anonymised Git only contains the functions used in their method*), which raises questions about their reproducibility. **Please include the code for your experiment in the final Git.**

---

> ### Author Rebuttal · Authors · 2026-03-31
>
> We thank the reviewer for noting that our paper tackles **the highly relevant problem** of making LLM-as-a-judge more reliable, a topic that **remains underexplored**, and for finding the paper **overall well written and structured** with **originality** as one of its strengths. We address each of the reviewer's concerns below.
>
> ---
>
> # Q1. Did you analyse where contemporary LLM-as-a-Judge approaches are located?
>
> To address this, we analyze where contemporary LLM-as-a-Judge models lie within the parameter regime characterized in our paper, and whether they satisfy the requirements in Proposition 5.1.
>
> Following the Chatbot Arena experiments in Section 6.2, we estimate each judge’s $(q_0, q_1)$ using pairwise comparison data, where ground truth is defined by human preferences over Alpaca-13B outputs. We then map these estimates to identify when LLM–based evaluation $\hat{\theta}$ achieves lower variance than direct human evaluation $\hat{\phi}$. The figure visualizes this by showing the $(q_0, q_1)$ space, where color indicates the range of $\theta$ over which LLM-as-a-Judge is more variance-efficient.
>
> **Figure A:** https://anonymous.4open.science/r/LLM-judge-reporting/figs/current.png
>
> Our analysis shows that current LLM judges do not yet satisfy the requirements of Proposition 5.1. While stronger models achieve higher $(q_0, q_1)$ values, they remain outside the region where the condition holds, so human evaluation still has a variance advantage.
>
> At the same time, we observe a consistent trend. As model performance improves, $(q_0, q_1)$ moves closer to the favorable regime. For example, within the Claude series, performance improves from Haiku 4.5 to Sonnet 4.6 (and Opus 4.6), with the latter models lying closer to the advantageous region.
>
> Taken together, current LLM-as-a-Judge approaches do not yet meet the requirements of Proposition 5.1, but are steadily approaching them, suggesting that continued improvements may soon make LLM-as-a-Judge both more cost and variance efficient than human evaluation.
>
> ---
>
> # Q2. & W1. Can you provide the results of the experiment in Section 6.2 for at least one more LLM judge?
>
> In addition to GPT-4.1 mini, we conducted experiments with three additional LLM-as-a-judge models (GPT-4.1, Claude Haiku 4.5, and Gemini 3 Flash) while keeping all other experimental settings identical to those in the main paper. The results show that our method mitigates bias across different judge models. These results are also provided at https://anonymous.4open.science/r/LLM-judge-reporting/figs/different_judge/
>
> Furthermore, we clarify in the revised manuscript that our method requires estimating the sensitivity and specificity of each LLM-as-a-judge, reflecting its dependence on the underlying model.
>
> ---
>
> # Q3. & W4. Share the code.
>
> We apologize for only releasing partial code. We have now uploaded the complete code for all experiments at the anonymized link:
> https://anonymous.4open.science/r/LLM-judge-reporting/
>
> ---
>
> # W2. What if sensitivity != specificity for the analysis in Section 5?
>
> We extend Proposition 5.1 to the general case where sensitivity $q_0$ and specificity $q_1$ may differ. The interval in L309 is then given by
>
> $$
> \theta \in
> \left[
> \frac{1+\Delta}{2} -
> \sqrt{\frac{(1+\Delta)^2}{4} - \frac{ q_0(1-q_0)}{(q_0+q_1-1)^2}},
> \frac{1+\Delta}{2} +
> \sqrt{\frac{(1+\Delta)^2}{4} - \frac{ q_0(1-q_0)}{(q_0+q_1-1)^2}}
> \right],
> $$
> where we define $\Delta := \frac{q_0(1-q_0) - q_1(1-q_1)}{(q_0+q_1-1)^2}$.
>
> Additionally, we will include figures showing how the regime in which LLM-as-a-judge achieves lower variance than human-only evaluation changes as a function of $q_1$ (with $q_0$ fixed) and $q_0$ (with $q_1$ fixed), with the true accuracy $\theta$ on the y-axis, which is a generalized version of Figure 4.
>
> **Figure B:** https://anonymous.4open.science/r/LLM-judge-reporting/figs/general.png
>
> ---
>
> # W3. The results from Appendix F should be explicitly discussed in the main text.
>
> Our method is valid for any $(q_0, q_1)$ with $q_0+q_1>1$. The parameters in Section 6.1 were chosen to reflect a realistic scenario where sensitivity and specificity are unequal and both strictly less than 1. We will add the following after L316 to make this generalizability explicit:
>
> > The sensitivity and specificity values used here were chosen to illustrate a realistic scenario where the two are unequal. Across all choices of $(q_0, q_1)$ with $q_0 + q_1 > 1$, adaptive allocation consistently gives shorter confidence intervals than symmetric allocation, with the benefit becoming more pronounced as $\hat{p}$ departs from 0.5. Appendix F presents results across a broader range of $(q_0, q_1)$ values, confirming this pattern.

---

> > ### Author Rebuttal · Reviewer_EwMQ · 2026-04-01
> >
> > Thank you for your reply. You state in your **reply to Q1**:
> >
> > >Our analysis shows that current LLM judges do not yet satisfy the requirements of Proposition 5.1. While stronger models achieve higher values, they remain outside the region where the condition holds, so human evaluation still has a variance advantage.
> >
> > This sadly means, that your proposed method is not applicable to actual LLM-as-a-judge settings today, and it remains doubtful if LLMs will ever reach a performance where this is the case. Therefore, I cannot improve my scores, but would encourage further research on this to improve the method.
> >
> > The replies to the remaining weaknesses and questions are adequate.

---

> > > ### Author Response · Authors · 2026-04-02
> > >
> > > Thank you for your thoughtful feedback. While we agree that current LLM judges may not universally satisfy the condition in Proposition 5.1, we would like to emphasize two important points.
> > >
> > > First, the validity of the condition is inherently task-dependent. While no LLM-as-a-judge model appears to satisfy the condition in Chatbot Arena, it holds in other tasks. In particular, on the AlpacaEval benchmark [1], when evaluating GPT-4 win rates, we observe that an LLM-as-a-judge model satisfies the required sensitivity and specificity conditions, as shown in the figure below. This suggests that the condition is not fundamentally unattainable in practice, but rather depends on the difficulty and structure of the evaluation task.
> > >
> > > **Figure:** https://anonymous.4open.science/r/LLM-judge-reporting/figs/AlpacaEval.png
> > >
> > > Second, the condition is about the variance of the statistical estimator rather than the overall usefulness of LLM-based evaluation. Even when the condition is not satisfied, our method remains applicable and useful in practice, providing reliable estimates while significantly reducing evaluation cost compared to fully human-based assessment. Therefore, the practical benefits of leveraging LLM-as-a-judge remain intact regardless of whether the condition strictly holds.
> > >
> > > [1] Li et al., AlpacaEval: An Automatic Evaluator of Instruction-following Models, GitHub repository, 2023.

---

### Official Review · Reviewer_gMWz · 2026-02-24

**Soundness:** 1
**Presentation:** 2
**Significance:** 2
**Originality:** 1
**Overall Recommendation:** 1
**Confidence:** 3

**Summary:**

The authors take a bias correction method from the 70s in epidemiology, and suggest applying it to llm-as-a-judge evaluations. They demonstrate how to use it on chatbot arena.

**Compliance With Llm Reviewing Policy:**

Affirmed.

**Final Justification:**

While I appreciate the authors engaging with my comments, the responses and suggested changes to the paper do not justify changing my overall assessment.

**Key Questions For Authors:**

The subject is timely, could you extend your framework to cover newer, more recent (and more appropriate) methods of bias correction?

**Strengths And Weaknesses:**

Strengths: I appreciate the formalisation of this problem, trying to use statistical methods (e.g., uncertainty quantification) and applying it to real llm-as-a-judge data.

Weaknesses: the methodology is a very basic one (their citation is from 1978, almost 50 years ago). It assumes a true/false label and a ground truth. In LLM-as-a-Judge evals there is no ground truth (of course you can assume that humans are the ground truth), but there are usually multiple human raters and you should take this into account. Additionally there are multiple LLM-as-a-Judge and multiple repeats. We also know that there are different types of biases and this method does not at all separate between them (which is the more interesting question). Lastly, the Chatbot Arena is made of pairwise comparisons meaning that each sample is not independent (it depends what it was compared to). The 1978 formula was built for true/false binary classification s so is probably not applicable here. Similarly, the confidence interval formula also assumes independence. This type of data mixed-effects modeling.
In terms of presentation, Figures 1 and 2 (specifically 2a and 2c) shown well-known concepts. It would be much more interesting to use this space to show something novel.
The problem is relevant but there’s been lots of work on this in recent years, so would have been more interesting to build on these ones and take into account the complexity of llm evaluation data.

---

> ### Author Rebuttal · Authors · 2026-03-31
>
> We thank Reviewer gMWz for the detailed feedback. We hope the following responses clarify the contributions of our work.
>
> # W1. The methodology is a very basic one.
>
> We respectfully clarify that our paper extends beyond applying a classical estimator. While the bias correction formula serves as a known starting point, the application of this method to LLM-as-a-judge evaluation is novel and addresses a gap that is widely present yet consistently overlooked in practice, as evidenced by our Chatbot Arena experiments. Beyond this application, the core contributions are:
> - confidence intervals that jointly account for uncertainty from both the test and calibration datasets,
> - an adaptive calibration allocation strategy that minimizes interval width (Proposition 4.2),
> - a theoretical characterization of parameter regimes in which LLM-based evaluation is more reliable than human-only evaluation (Proposition 5.1), and
> - robustness under distribution shift between calibration and test sets, in contrast to existing approaches (Figure 7).
>
> We also share the assessment of Reviewer 4JMr, who notes that "identifying and solving a pertinent problem with an elegant and simple solution **should be considered a strength, and not a weakness**". Reviewer EwMQ similarly states that "originality can be seen as one of the **strengths of the paper**".
>
> ---
>
> # W2. The formula assumes independent samples, but samples in Chatbot Arena are not independent.
>
> We agree that, due to the pairwise comparison structure of Chatbot Arena, samples are not independent. However, as is common in many machine learning studies, the independence assumption is adopted to make theoretical analysis tractable and may not strictly hold in real-world benchmarks. Despite this gap, our empirical results show that the theoretically motivated method remains effective in practice. In particular, we observe that our method reduces bias toward zero in the Chatbot Arena results in Figure 6, suggesting that it is robust even when the independence assumption is violated.
>
> We will include this discussion in the revised manuscript.
>
> ---
>
> # W3. There is no ground truth in LLM-as-a-judge evaluations.
>
> We agree that in LLM-as-a-judge evaluations, there may not always be a definitive ground truth. We also agree that multiple sources of bias exist and that disentangling these biases is an interesting problem.
>
> However, our work focuses on providing an initial formalization of the problem of adjusting the bias in LLM-based judgment under imperfect LLM evaluators. As a pioneering work, we intentionally consider a simplified setting.
>
> As the reviewer points out, extending this framework to more realistic scenarios would require explicitly modeling additional factors. For example, when multiple LLM judges are used, our framework can be applied to each judge individually by estimating judge-specific sensitivity and specificity, and then aggregating the corrected estimates, as discussed in the future work section of the submitted manuscript. This approach allows us to account for judge-specific biases.
>
> Rather than incorporating all these complexities in real-world cases, this paper aims to take a first step toward the formulation and analysis of LLM-as-a-Judge. We therefore consider the extensions discussed above to be important directions for future work.
>
> ---
>
> # W4. & Q1. Could you extend your framework to cover newer and more recent methods of bias correction?
>
> We acknowledge the growing body of work investigating the origins of bias in LLMs and proposing debiasing strategies that do not rely on calibration datasets with ground-truth labels. Our framework and these newer methods (e.g., prompt-based debiasing, fine-tuning) address fundamentally different layers of the problem. The latter aim to improve the judge itself, while our method corrects for any residual bias in the final evaluation score. These two approaches are complementary rather than competing. Even when a newer debiasing technique is applied to the judge, our framework remains applicable and benefits from the improved sensitivity and specificity. Moreover, our method offers guarantees that most existing debiasing approaches do not provide valid confidence intervals and robustness under distribution shift. Until inherently unbiased judges demonstrate consistent robustness across diverse settings, our method provides a practical and theoretically grounded alternative. We will discuss this complementary relationship more explicitly in the revised manuscript.

---

> > ### Author Rebuttal · Reviewer_gMWz · 2026-04-02
> >
> > I thank the authors for their rebuttal. While I acknowledge the contributions beyond the 1978 formula, my core concerns remain insufficiently addressed.
> >
> > On independence (W2): The authors acknowledge the violation in Chatbot Arena but offer no analysis of its consequences for coverage. For a paper advocating statistical rigour in reporting, stating that "many ML studies adopt this assumption" is not a sufficient justification.
> >
> > On scope and modeling limitations (W1, W3, W4): My central concern remains that the binary i.i.d. single-judge framework is too restrictive for the generality claimed by the title. There are well-known statistical methods that address these limitations. For instance, GLMs naturally handle ordinal scores, pairwise comparisons, multiple human and LLM graders, bias decomposition (e.g., self-bias, length bias), structured dependencies, and provide principled uncertainty quantification. All of this within a unified framework and with fewer violated assumptions.
> > I do recognise that the reviewed paper's distribution shift analysis and computational simplicity offer value, but the authors should scope their claims more precisely to settings where their assumptions hold (e.g., binary verifiable tasks, i.i.d. samples), rather than presenting this as a general solution for LLM-as-a-judge evaluation. Overall, there are more appropriate ways to model LLM-as-a-judge evaluation that are both feasible and practical.
> >
> > While I appreciate the authors engaging with my comments, the responses and suggested changes to the paper do not justify changing my overall assessment.

---

> > > ### Author Response · Authors · 2026-04-03
> > >
> > > We thank the reviewer for the additional comments and take this opportunity to briefly clarify our responses.
> > >
> > > Regarding independence, we agree that in general pairwise comparisons may not be independent, especially if the same LLM response is compared over multiple pairwise comparisons. However in our setting, they are independent as we fix one LLM as target of our estimator for its win rate and **randomly select the opponent**. Crucially we do not reuse the same LLM response across multiple pairwise comparisons. At the same time, we note that (i) our experiments on Chatbot Arena show that the proposed correction **remains effective in practice**, **including empirical coverage** of our confidence intervals remaining near nominal, as shown in Table 1 of our manuscript; (ii) many commonly used statistical models (**including GLMs** [1]) also rely on **independence assumptions**; and (iii) we have already outlined how to extend our method to **explicitly model such dependencies** (e.g., length bias) in the future work section of our manuscript.
> > >
> > > On modeling scope, the binary setting reflects common LLM-as-a-judge practice based on pairwise comparisons, where our framework directly applies. We further show that our method influences decision-making by **recovering correct model rankings**, see response to Reviewer 4JMr `W2.`. Moreover, our method naturally extends to **multinomial settings**, as outlined in the future work section, and we provide additional experimental evidence in our response to Reviewer YUnH `W1.`. We will revise the introduction to more clearly specify this scope.
> > >
> > > [1] As noted in McCullagh and Nelder (Generalized Linear Models, 1989, p. 26), in GLMs, "a vector of observations is assumed to be a realization of a random variable whose components are independently distributed".

---

### Official Review · Reviewer_4JMr · 2026-03-10

**Soundness:** 3
**Presentation:** 3
**Significance:** 4
**Originality:** 3
**Overall Recommendation:** 5
**Confidence:** 4

**Summary:**

The authors provide a mathematically sound framework for correcting the bias in LLM-as-a-judge evaluations by, naturally, calibrating for the performance of the judge. They provide estimates/quantification for the confidence intervals of the proposed method, and examine distribution shifts between calibration and test sets. The paper overall presents a simple pipeline to increase the reliability of LLM-as-a-judge evaluations, a pertinent setting in the community.

**Compliance With Llm Reviewing Policy:**

Affirmed.

**Final Justification:**

The authors addressed all my concerns, improving my assessment of the paper.

**Key Questions For Authors:**

- I did not see any mention of human disagreement in the paper. Preference data, like all "subjective" domains, have been shown to contain a certain degree of disagreement between annotators, which does not necessarily arise from errors. How do you view your method within that context, and do/could the presented methods account for diverse perspectives in the calibration set?

**Limitations:**

- Stemming from my question, the authors have silently assumed that there is a ground truth which humans can unequivocally provide. There is no discussion of the influence of human errors and human disagreement, the latter of which might stem from a genuine disagreement between individuals rather than error.

**Strengths And Weaknesses:**

- Soundness: While I haven't checked all the derivations in detail, the approach and results are sound. The motivation makes intuitive sense, and the results are in line with expectations given the setup. The authors also link to an anonymous repo, providing some guarantees that the adoption and impact of this work, as well as future scrutiny and improvements. Despite the theoretical soundness, there is little empirical validation. Notably, there is only one experiment with real data, and the judge used is relatively obsolete given the pace of progress. This raises real questions about whether the additional effort and resources spent on a calibration set and setting up the pipeline to account for it really influence the results. Critically, there is no direct demonstration of "flipped" results with and without calibration, which to my understanding was the basic premise of the paper; we do not necessarily care about the absolute value of the evaluation metrics, which is what bias directly influences, but rather the relative performance of different models. That is to say, does the additional bias introduced by the uncalibrated metrics actually influence decision-making? How often and where has it happened? A plausible alternative, weakly supported by your results, is that all models are biased in similar ways consistently, potentially resulting in no additional practical benefits from using this method. I'm using this as an example to demonstrate that empirical results showing the pressing need for such a method would greatly increase the impact of and the trust in the paper. Additionally, since you are proposing collecting preference data, it may be useful to show DPO or related results for the judge: why not use annotated data to train the judge? I understand this slightly departs from the goal of this work, but asking practitioners to collect data then begs the question.
- Presentation: The presentation is mostly clear. There are a few places within the paper where the reader is left wondering without reading the cited references. For me, the most prominent example is the first paragraph of the introduction, where a lot is essentially left as an exercise to the reader. Related work is also placed in the appendix, which would have otherwise helped to contextualize the authors contributions.
- Significance: Given the prevalence of LLM-as-a-judge frameworks, addressing the unreliability of judge evaluations is of major importance in the community.
- Originality: The authors leverage known methods to address the problem, but I believe that shouldn't subtract from the importance of the problem they are trying to solve. Identifying and solving a pertinent problem with an elegant and simple solution should be considered a strength, and not a weakness.

---

> ### Author Rebuttal · Authors · 2026-03-31
>
> We thank the reviewer for finding our approach and results **sound**, with the motivation **making intuitive sense** and for noting that **identifying and solving a pertinent problem** with an **elegant and simple solution** should be considered a strength given the **major importance** of addressing unreliability in LLM-as-a-judge evaluations. We address each of the reviewer's concerns below.
>
> ---
>
> # Q1. I did not see any mention of human disagreement in the paper. How do you view your method within that context, and do/could the presented methods account for diverse perspectives in the calibration set?
>
> We thank the reviewer for raising this point. We agree that human disagreement is an important aspect of LLM-as-a-judge evaluations, especially when labels reflect inherently diverse perspectives rather than a single objective ground truth.
>
> In this work, our formulation assumes a simplified setting in which the calibration set provides consistent labels. This assumption allows us to focus on a first-step formalization, and as a result, human disagreement is not explicitly modeled in the current framework. We will clarify these assumptions in the revised manuscript.
>
> Nevertheless, as briefly discussed in the future work section, our method can be extended to account for such disagreement. For example, when annotators can be grouped into sets with internally consistent preferences, one can construct group-specific calibration sets and aggregate the resulting estimates, allowing the framework to capture diverse perspectives.
>
> ---
>
> # W1. The judge used is relatively obsolete.
>
> In addition to GPT-4.1 mini, we conducted experiments with three additional LLM-as-a-judge models (GPT-4.1, Claude Haiku 4.5, and Gemini 3 Flash) while keeping all other experimental settings identical to those in the main paper. The results show that our method mitigates bias across different judge models. These results are also provided at https://anonymous.4open.science/r/LLM-judge-reporting/figs/different_judge/
>
> ---
>
> # W2. There is no direct demonstration of flipped results.
>
> To show that uncalibrated LLM judges can flip rankings and that calibration corrects this, we evaluate six models on Chatbot Arena using three LLM judges (same setup). The ground-truth ranking is: `GPT-4 (0.850) > Claude-v1 (0.786) > Vicuna-13B (0.621) > Alpaca-13B (0.358) > FastChat-T5-3B (0.300) > LLaMA-13B (0.209)`.
>
> We report Kendall-τ and Rank Match (RM; exact ranking recovery) over 100 seeds.
>
> **Table A: Gemini 3 Flash**
> ||τ|RM|
> |-|-|-|
> |Raw|0.867|0%|
> |Adj|0.891|37%|
>
> **Table B: Claude Haiku 4.5**
> ||τ|RM|
> |-|-|-|
> |Raw|0.876|7%|
> |Adj|0.900|40%|
>
> **Table C: GPT-4.1**
> ||τ|RM|
> |-|-|-|
> |Raw|0.981|86%|
> |Adj|0.915|46%|
>
> **Table D: GPT-4.1 (more calib., less test)**
> ||τ|RM|
> |-|-|-|
> |Raw|0.948|61%|
> |Adj|0.967|75%|
>
> Table A shows a ranking flip in all seeds (RM=0%), so the raw judge consistently selects the wrong model, but calibration fixes this. Table B shows the same failure mode.
>
> In contrast, Table C shows that for GPT-4.1 the uncalibrated ranking is already largely correct. In this regime, calibration can introduce additional variance due to limited calibration data, which may slightly degrade ranking metrics. Table D shows this is not fundamental: increasing the calibration set size improves ordering. The lower Raw scores in Table D are due to the reduced test set, as more data is allocated to calibration.
>
> In summary, uncalibrated LLM judges not only shift scores but alter rankings, leading to wrong model selection. Calibration removes this and restores correct ordering.
>
> ---
>
> # W3. It may be useful to show DPO or related results for the judge.
>
> Thanks for the great suggestion. We agree that the collected preference data could be used to train or improve a judge model using DPO or related methods.
>
> However, our primary focus is to improve the LLM-based judgment for a frozen LLM, instead of refining the LLM itself, which seems to be an orthogonal direction. We will leave it as a future work and discuss such possibility in the revised manuscript.
>
> ---
>
> # W4. The reader is left wondering without redeaing the cited references.
>
> Thank you for pointing this out. We will revise the first paragraph of the introduction to make it more self-contained and move the related work section from the appendix to the main text. If there are specific parts you found unclear, we would appreciate further pointers and will address them while improving the overall clarity of the paper.

---

> > ### Author Rebuttal · Reviewer_4JMr · 2026-03-31
> >
> > The authors have reasonably addressed my concerns and shared new results to cover for the gaps of the original submission.

---

### Official Review · Reviewer_YUnH · 2026-03-13

**Soundness:** 4
**Presentation:** 4
**Significance:** 3
**Originality:** 3
**Overall Recommendation:** 5
**Confidence:** 4

**Summary:**

This paper points out that the naive estimator of accuracy ($\hat{p}$), typically reported in LLM-as-a-judge, is biased.
Then, it proposes a method to correct the bias in the naive estimator.
The proposed method estimates sensitivity and specificity from a small amount of human-annotated calibration data and then adjusts the naive estimator using sensitivity and specificity.
The authors also derive the confidence interval of the proposed estimator and the optimal allocation of calibration samples to shorten the confidence interval.
Furthermore, they analyze the conditions under which the proposed method works more effectively compared to evaluation using a small amount of human-annotated data.
Experiments using both synthetic and real data demonstrate the effectiveness of the proposed method.

**Compliance With Llm Reviewing Policy:**

Affirmed.

**Final Justification:**

I read the authors' responses and I keep my score 5 (acceptance).

**Key Questions For Authors:**

- Have you conducted experiments with different Judge models?
- Does the proposed estimator become unstable when $q_0 + q_1 - 1$ is small? Does this cause problems?
- While existing methods may be biased under distribution shifts, might they offer an advantage in terms of variance when $P \simeq Q$?
- Human labels are treated as ground truth, but in real-world scenarios, disagreements between humans can occur. Can the proposed method still be applicable in such cases?

**Limitations:**

yes

**Strengths And Weaknesses:**

## Strength
- The problem formulation and approach are simple and clear.
- The paper proposes not only an unbiased estimator but also addresses various associated challenges. First, it provides confidence intervals that reflect the uncertainty in both the test samples and calibration samples. Second, it derives the optimal calibration set allocation that minimizes the confidence intervals. Third, it provides a theoretical analysis comparing LLM-as-a-judge using the proposed method with human-only evaluation.
- The comparison with existing correction methods, organized from the perspective of robustness against distribution shifts, is interesting.
- The experiments clearly demonstrate the effectiveness of the proposed method.

## Weakness
The experimental verification is limited:
- The mathematical and experimental setup relies on binary judgments, and its effectiveness for more general multi-value or ranking evaluations is not directly demonstrated.
- Experiments with real data are limited to Chatbot Arena and GPT-4.1-mini judges, and the generalizability when judge models or tasks change has not yet been sufficiently verified.
- Comparisons with existing methods for improving LLM-as-a-judge accuracy have not been conducted.

---

> ### Author Rebuttal · Authors · 2026-03-31
>
> We appreciate the reviewer for finding our problem formulation **simple and clear** and for recognizing that the paper **addresses** not only bias correction but also **various associated challenges** including confidence intervals, optimal allocation, and theoretical comparison with human-only evaluation. We address each of the reviewer's concerns below.
>
> ---
>
> # Q1. & W2. Experiments with different judge models.
>
> In addition to GPT-4.1 mini, we conducted experiments with three additional LLM-as-a-judge models (GPT-4.1, Claude Haiku 4.5, and Gemini 3 Flash) while keeping all other experimental settings identical to those in the main paper. The results show that our method mitigates bias across different judge models. These results are also provided at https://anonymous.4open.science/r/LLM-judge-reporting/figs/different_judge/
>
> ---
>
> # Q2. What if $q_0+q_1-1$ is small?
>
> When $q_0+q_1-1$ is small, our estimator becomes unstable. However, this issue is not specific to our estimator, but applies to any estimator based on LLM-as-a-judge. To illustrate this, consider the extreme case where $q_0=q_1=0.5$, so that $q_0+q_1=0$. In this setting, since $q_0=q_1=0.5$, the LLM outputs true or false at random regardless of the input response pairs. Therefore, any estimator based on LLM-as-a-judge scores reduces to random guessing.
>
> ---
>
> # Q3. Do existing methods provide a variance advantage under no distribution shift $P=Q$?
>
> Yes, when $P=Q$, existing methods such as the difference estimator in Table 3 can achieve lower variance than ours in some cases, as shown in prior work described in L97.
>
> However, our method is designed to remain valid under distribution shifts, trading off some variance for improved robustness, which we believe better reflects realistic evaluation scenarios. In practice, the assumption $P=Q$ often does not hold in LLM-as-a-judge settings, as noted in L367. Moreover, under the assumption $P=Q$, the target score can be directly estimated from the calibration dataset without using an LLM-as-a-judge, as discussed in L375.
>
> ---
>
> # Q4. Can the proposed method handle disagreement among human annotators?
>
> Due to the response length limit, please refer to our response to Reviewer 4JMr’s `Q1`.
>
> ---
>
> # W1. More general multi-value or ranking evaluations.
>
> While our paper focuses on binary judgments, the framework naturally extends to the multinomial setting, as discussed in the second point of the future work section. We demonstrate this with a synthetic experiment modeled after the conference paper review scale, which uses $K=6$ ordered categories: Strong Reject (1), Reject (2), Weak Reject (3), Weak Accept (4), Accept (5), and Strong Accept (6).
>
> We generate $n=1000$ submissions, each evaluated by 4 LLM judges. Judge scores are drawn from a confusion matrix $M$ with $M_{a,b} = \Pr(\hat{Z}=a \mid Z=b)$, parameterized to exhibit central tendency bias. A calibration set of $m=200$ labeled submissions is used to estimate $\hat{M}$ (the multinomial analogue of sensitivity and specificity), yielding the biased-adjusted estimator $\hat{\theta} = \hat{M}^{-1}\hat{p}$. Results are averaged over 10 Monte Carlo trials.
>
> As shown in Table, the naive estimator $\hat{p}$ overestimates the middle categories (Weak Reject, Weak Accept) and underestimates the tails (Strong Reject, Strong Accept). The biased-adjusted estimator $\hat{\theta}$ recovers the ground-truth counts with reduced bias across all categories.
>
> **Table.** Estimated number of submissions per recommendation score. Entries show Mean (SE) over 10 Monte Carlo repetitions.
>
> |  | Strong Reject | Reject | Weak Reject | Weak Accept | Accept | Strong Accept |
> |:---|:---:|:---:|:---:|:---:|:---:|:---:|
> | Ground truth | 50 | 150 | 300 | 250 | 200 | 50 |
> | $\hat{p}$ | 39.8 (1.9) | 127.1 (2.3) | 321.6 (3.4) | 311.7 (2.4) | 156.5 (2.1) | 43.2 (1.1) |
> | $\hat{\theta}$ | 53.7 (3.8) | 149.9 (6.0) | 279.9 (11.8) | 264.5 (11.8) | 195.0 (7.1) | 57.0 (3.7) |
>
> While we provide additional experiments for multinomial (multi-value) settings, extending the framework to ranking scenarios is also a promising direction. Although ordinal multinomial labels capture coarse ranking information, full ranking problems require modeling relative preferences across items, which we leave for future work.
>
> ---
>
> # W3. Comparisons with existing methods for improving LLM-as-a-judge accuracy.
>
> Our method provides a practical and reliable adjustment for real-world evaluation under unavoidable bias in LLM-as-a-judge settings. While prior work has focused on reducing or eliminating bias in the judge itself, these approaches are orthogonal to our goal of maintaining reliable evaluations despite the presence of bias. Achieving inherently unbiased LLMs remains an important long-term objective, but such methods have yet to demonstrate consistent robustness across diverse real-world scenarios, highlighting the need for complementary approaches such as ours.

---

> > ### Author Rebuttal · Reviewer_YUnH · 2026-04-04
> >
> > I thank the authors for their responses. The authors have addressed my concerns.

---

### Decision · Program_Chairs · 2026-04-30

**Decision:**

Accept (regular)

**Comment:**

This paper proposes a calibration-based framework for LLM-as-a-judge evaluation that assumes access to a human-labeled calibration set treated as ground truth, and uses it to correct bias from imperfect judge sensitivity and specificity while also providing confidence intervals and calibration analysis.

Reviewers highlighted the importance, clear formulation, solid technical contribution, and practical relevance. The rebuttal strengthened the empirical case by adding additional judge experiments and ranking analyses.

Remaining concerns from the positive reviewers were mainly about limited empirical breadth, the need to better integrate the new rebuttal results, and the fact that some of the favorable variance regimes may not yet hold broadly for current judges.

Reviewer gMWz gave score 1, and argued that human true labels, binary single-judge i.i.d. modeling, and independence in pairwise evaluation settings are too strong, and that richer statistical approaches may better capture real LLM-as-a-judge settings. While I agree these are meaningful weaknesses, these can be discussed as limitations in the paper and the community may still be able to benefit from the proposed ideas in the paper. I therefore support acceptance, while strongly encouraging the authors in the camera-ready version to explicitly discuss the assumptions and weaknesses emphasized by Reviewer gMWz.